# An Analysis of Constant Step Size SGD in the Non-convex Regime: Asymptotic Normality and Bias

**Lu Yu**
University of Toronto & Vector Institute
stat.yu@mail.utoronto.ca

**Krishnakumar Balasubramanian**
University of California, Davis
kbala@ucdavis.edu

**Stanislav Volgushev**
University of Toronto
stanislav.volgushev@utoronto.ca

**Murat A. Erdogdu**
University of Toronto & Vector Institute
erdogdu@cs.toronto.edu

## Abstract

Structured non-convex learning problems, for which critical points have favorable statistical properties, arise frequently in statistical machine learning. Algorithmic convergence and statistical estimation rates are well-understood for such problems. However, quantifying the uncertainty associated with the underlying training algorithm is not well-studied in the non-convex setting. In order to address this short-coming, in this work, we establish an asymptotic normality result for the constant step size stochastic gradient descent (SGD) algorithm—a widely used algorithm in practice. Specifically, based on the relationship between SGD and Markov Chains [1], we show that the average of SGD iterates is asymptotically normally distributed around the expected value of their unique invariant distribution, as long as the non-convex and non-smooth objective function satisfies a dissipativity property. We also characterize the bias between this expected value and the critical points of the objective function under various local regularity conditions. Together, the above two results could be leveraged to construct confidence intervals for non-convex problems that are trained using the SGD algorithm.

## 1 Introduction

Non-convex learning problems are prevalent in modern statistical machine learning applications such as matrix and tensor decomposition [2, 3, 4, 5, 6, 7, 8], deep neural networks [9, 10, 11], and robust empirical risk minimization [12, 13, 14]. Developing theoretically principled approaches for tackling such non-convex problems depends critically on the interplay between two aspects. From a computational perspective, variants of stochastic gradient descent (SGD) converge to first-order critical points [15, 16] or local minimizers [17, 2, 18, 19] of the objective function. From a statistical perspective, *oftentimes* these critical points or local minimizers have nice statistical properties [20, 3, 12, 21, 22, 5]; see also [23] for a counterexample. For the purpose of uncertainty quantification in such non-convex settings, studying the fluctuations of iterative algorithms used for training becomes extremely important. In this work, we focus on the widely used constant step size SGD, and develop results for quantifying the uncertainty associated with this algorithm for a class of non-convex problems.

We consider minimizing a non-smooth and non-convex objective function $f \colon \mathbb{R}^d \to \mathbb{R}$,

$$\min_{\theta \in \mathbb{R}^d} f(\theta) \,. \tag{1}$$

35th Conference on Neural Information Processing Systems (NeurIPS 2021).

The iterations of SGD with a constant step size $\eta > 0$, initialized at $\theta_0^{(\eta)} \equiv \theta_0 \in \mathbb{R}^d$, are given by

$$\theta_{k+1}^{(\eta)} = \theta_k^{(\eta)} - \eta\big(\nabla f(\theta_k^{(\eta)}) + \xi_{k+1}(\theta_k^{(\eta)})\big), \quad k \geq 0, \tag{2}$$

where $\{\xi_k\}_{k \geq 1}$ is a sequence of random functions from $\mathbb{R}^d$ to $\mathbb{R}^d$ corresponding to the stochasticity in the gradient estimate. Several problems in machine learning and statistics are naturally formulated as the optimization problem in (1), where the function $f(\theta)$ is given by

$$f(\theta) \coloneqq \int F(\theta, Z)\, dP(Z), \tag{3}$$

for an unknown distribution over the random variable $Z \in \mathbb{R}^p$. The function $F(\theta, Z)$ is typically the loss function composed with functions from the hypothesis class parametrized by $\theta \in \mathbb{R}^d$. In online SGD with batch size $b$, at each iteration $k$, $b$ independent samples $Z_j \sim P(Z)$ are used to estimate the true gradient with $\frac{1}{b} \sum_{j=1}^{b} \nabla F(\theta_k^{(\eta)}, Z_j)$. The above iterates are indeed a special case of the iterates in (2), with the noise sequence $\{\xi_{k+1}(\theta_k^{(\eta)})\}_{k \geq 0}$ given by

$$\xi_{k+1}(\theta_k^{(\eta)}) \coloneqq \frac{1}{b} \sum_{j=1}^{b} \left[ \nabla F(\theta_k^{(\eta)}, Z_j) - \nabla f(\theta_k^{(\eta)}) \right]. \tag{4}$$

Although proposed in the 1950s by [24], SGD has been the algorithm of choice for training statistical models due to its simplicity, and superior performance in large-scale settings [25, 1, 26, 27]. However, the fluctuations of this algorithm is well-understood only when the objective function $f$ is strongly convex and smooth, and the step size $\eta$ satisfies a specific decreasing schedule so that the iterates asymptotically converge to the *unique* minimizer [28, 29, 30]. On the other hand, it is well-known that the SGD iterates in (2) can be viewed as a Markov chain which allows them to converge to a random vector rather than a single critical point [1]. Building on this analogy between SGD and Markov chains, the aforementioned shortcomings can be alleviated by simply relaxing the global smoothness as well as the strong convexity assumptions to the tails of the objective function $f$, which allows for a flexible non-convex structure around the region of interest. Similar kinds of tail relaxations have been successfully employed in the diffusion theory when the target potential is non-convex [31, 32, 33], but they are not studied in the context of non-convex optimization when the algorithm is SGD. In this work, we study the fluctuations and the bias of the averaged SGD iterates in (2), around the first-order critical points of the minimization problem (1). Our contributions can be summarized as follows.

- For a non-convex and non-smooth objective function $f$ with tails growing at least quadratically, we establish the uniqueness of the stationary distribution of the constant step size SGD iterates in Proposition 2.1, and the asymptotic normality of Polyak-Ruppert averaging in Theorem 2.1. To the best of our knowledge, these are the first uniqueness and normality results for the SGD algorithm when the objective function is non-convex (even not strongly convex) and non-smooth.
- We further show in Theorems 3.1 and 3.2 that, with additional local smoothness assumptions on the non-convex objective function $f$, we can establish a control over the bias in terms of the step size. We further characterize the bias when the objective is (not strongly) convex in Theorem 3.3, providing a thorough bias analysis for the constant step size SGD under various settings that are frequently encountered in statistical learning.

Our results provide algorithm-dependent guarantees for uncertainty quantification, and they could be leveraged to obtain confidence intervals (CIs) for non-convex and non-smooth learning problems. This is contrary to the majority of the existing results in statistics, which only establish normality results for the true stationary point of the non-convex objective function; see for example [12, 34]. While being useful, such results completely ignore the computational hardships associated with non-convex optimization; hence, their practical implications are limited. On the other hand, in the optimization and learning theory literature, a majority of the existing results establish the rate of convergence of an algorithm to a critical point, and do not quantify the fluctuations associated with that algorithm. Our work bridges these separate lines of thought by providing asymptotic normality results directly for the SGD algorithm used for minimizing non-convex and non-smooth functions.

**More Related Works.** Establishing asymptotic normality results for the SGD algorithm began with the works of [35, 36, 37, 38, 39], with [28] providing a definitive result for strongly convex objectives. In particular, [28] and [38] established that the averaged SGD iterates with an appropriately chosen decreasing step size is asymptotically normal with the variance achieving the Cramer-Rao lower

bound for parameter estimation. Recent works, for example [40, 41, 29, 42, 43, 44], leverage the asymptotic normality analysis of [28], which leads to computing CIs for SGD. The benefits of constant step size SGD for faster convergence under overparametrization has also been demonstrated in the works of [45, 46, 47, 48]. The use of Markov chain theory to study constant step size stochastic approximation algorithms has been considered in several works [49, 50, 51, 25, 52, 53]. Recently, [1, 54] investigated the asymptotic variance of the constant step size SGD. We emphasize here that most of the above works assume strongly convex and smooth objective functions.

The non-linear autoregressive (NLAR) process [55, 56, 57] is a specification of our general framework (2) with the noise sequence $\{\xi_k\}_{k \geq 1}$ being a collection of i.i.d mean-zero random vectors with continuous density supported on $\mathbb{R}^d$. However, the methodology for establishing the geometric ergodicity of NLAR [55, 56, 57, 58] is by no means straightforward to carry over to the optimization setting, and does not generalize immediately to the state-dependent noise setup considered in our paper (see Assumption 2.3). In contrast, we establish the geometric ergodicity under easily verifiable assumptions on the objective function using tools from Markov chain theory. Moreover, additional steps are needed to go from geometric ergodicity to CLT results (especially if the chain starts with an arbitrary initial distribution), while we directly obtain a CLT result by explicitly leveraging the Markov chain structure.

Furthermore, there exists a vast literature on analyzing Langevin diffusion-based sampling algorithms which relies on the much simpler i.i.d. Gaussian noise sequence. We refer the interested reader to [59, 60, 32, 61, 62, 63, 64, 65, 66, 67, 68, 69, 70] and the references therein, for details.

Finally, recent work [71] establish the trade-off between the stability, statistical accuracy, and computational efficiency for the non-convex optimization algorithms.

**Notation.** For $a, b \in \mathbb{R}$, denote by $a \vee b$ and $a \wedge b$ the maximum and the minimum of $a$ and $b$, respectively. We use $\| \cdot \|$ to denote the Euclidean norm in $\mathbb{R}^d$. We denote the largest eigenvalue of the matrix $A$ as $\lambda_{\max}(A)$, and the smallest one as $\lambda_{\min}(A)$. Let $(\Omega, \mathcal{F}, \mathbb{P})$ represent a probability space, and denote by $\mathcal{B}(\mathbb{R}^d)$, the Borel $\sigma$-field of $\mathbb{R}^d$. Let $\mathcal{P}_k(\mathbb{R}^d) := \{\nu : \int_{\mathbb{R}^d} \|\theta\|^k \nu(d\theta) < \infty\}$ denote the set of probability measures with finite $k$-th moments. For a probability distribution $\pi$ and a function $g$ on $\mathcal{X}$, we define $\pi(g) := \int_{\mathcal{X}} g(x) d\pi(x)$, and $\mathcal{L}_2(\pi) := \{g : \mathcal{X} \to \mathbb{R} : \pi(g^2) < \infty\}$.

## 2 Central Limit Theorem for The Constant Step Size SGD

In this section, we establish an asymptotic central limit theorem (CLT) for the Polyak-Ruppert averaging of the constant step size SGD iterates given in (2) when the objective function is potentially non-convex, non-smooth, and has quadratically growing tails. More specifically, we first prove that there exists a unique stationary distribution $\pi_\eta \in \mathcal{P}_2(\mathbb{R}^d)$ for the Markov chain defined by the SGD algorithm when the objective function is dissipative (see Assumption 2.2) with gradient exhibiting at most linear growth (see Assumption 2.1). Furthermore, under the same conditions, we prove that a CLT holds for the Polyak-Ruppert averaging, and it is independent of the initialization. In what follows, we list and discuss the main assumptions required to establish a CLT for the SGD iterates, and compare them to those existing in the literature.

**Assumption 2.1** (Linear growth). *The gradient of the objective function $f$ has at most linear growth. That is, for some $L \geq 0$, we have $\|\nabla f(\theta)\| \leq L(1 + \|\theta\|)$ for all $\theta \in \mathbb{R}^d$.*

Majority of the results on SGD focus on smooth functions with gradients satisfying $\|\nabla f(\theta) - \nabla f(\theta')\| \leq \|\theta - \theta'\|$ for all $\theta, \theta' \in \mathbb{R}^d$; see e.g. [28, 1]. The above condition allows for non-smooth objectives, and is a significant relaxation of the standard Lipschitz gradient condition.

**Assumption 2.2** (Dissipativity). *The objective function $f$ is $(\alpha, \beta)$-dissipative. That is, there exists positive constants $\alpha, \beta$ such that $\langle \theta, \nabla f(\theta) \rangle \geq \alpha \|\theta\|^2 - \beta$ for all $\theta \in \mathbb{R}^d$.*

The dissipativity assumption has its origins in the analysis of dynamical systems, and is used widely in the analysis of optimization and learning algorithms [72, 31, 33, 73]. It also recently have been used in Bayesian analysis [74]. This assumption could be viewed as a relaxation of strong convexity since it restricts the quadratic growth assumption to the tails of the function $f$, enforcing no local growth around the first-order critical points. A canonical example for this condition is the sum of a quadratic and any non-convex function with bounded gradient. For example, consider the function $x \to x^2 + 10 \sin(x)$ which is clearly non-convex and $(1, 25)$-dissipative. It is worth mentioning

that many non-convex problems that arise in statistical learning such as phase retrieval [53] satisfy Assumption 2.2. We provide examples in Section 4.

**Assumption 2.3** (Noise sequence). *Gradient noise sequence $\{\xi_k\}_{k\geq 1}$ is a collection of i.i.d. random fields satisfying $\mathbb{E}[\xi_1(\theta)] = 0$ and $\mathbb{E}^{1/2}[\|\xi_1(\theta)\|^2] \leq L_\xi(1 + \|\theta\|)$, for any $\theta \in \mathbb{R}^d$ and a positive constant $L_\xi$. Moreover, for each $\theta \in \mathbb{R}^d$ the distribution of the random variable $\xi_1(\theta)$ can be decomposed as $\mu_{1,\theta} + \mu_{2,\theta}$ where $\mu_{1,\theta}$ has a density, say $p_\theta$, with respect to Lebesgue measure which satisfies $\inf_{\theta \in C} p_\theta(t) > 0$ for any bounded set $C$ and any $t \in \mathbb{R}^d$.*

Assumption 2.3 as formulated above is stronger than what is needed in the proofs. It can easily be seen that the lower bound on the density $p_\theta$ is only required to hold for a specific set whose form depends on $\eta$ and various constants from Assumptions 2.1–2.3. The form of this set is complicated, and an exact expression is given in the Appendix – see (12). We also emphasize that Assumption 2.3 does not specify any explicit parametric form for the distribution of the noise sequence contrary to recent works in non-convex settings where dissipitavity condition has been heavily utilized [31, 73, 33].

We now establish the existence and uniqueness of the stationary distribution of the SGD iterates.

**Proposition 2.1** (Ergodicity of SGD). *Let the Assumptions 2.1-2.3 hold, and the step size satisfy*

$$0 < \eta < \frac{\alpha - \sqrt{(\alpha^2 - (3L^2 + L_\xi))\vee 0}}{3L^2 + L_\xi}.$$

*(a) SGD (2) admits a unique stationary distribution $\pi_\eta \in \mathcal{P}_2(\mathbb{R}^d)$, depending on the step size $\eta$.*

*(b) For a test function $\phi : \mathbb{R}^d \to \mathbb{R}$ satisfying $|\phi(\theta)| \leq L_\phi(1 + \|\theta\|)$, $\forall \theta \in \mathbb{R}^d$ and some $L_\phi > 0$, and for any initialization $\theta_0^{(\eta)} = \theta_0 \in \mathbb{R}^d$ of the SGD algorithm, there exists $\rho \in (0, 1)$ and $\kappa$ (both depending on $\eta$) such that we have*

$$\left|\mathbb{E}\big[\phi\big(\theta_k^{(\eta)}\big)\big] - \pi_\eta(\phi)\right| \leq \kappa\,\rho^k(1 + \|\theta_0\|^2)\,, \quad \text{where } \pi_\eta(\phi) := \int \phi(x)d\pi_\eta(x).$$

The uniqueness of the stationary distribution of the constant step size SGD has been established in [1] for strongly convex and smooth objectives. In Proposition 2.1, we relax both of these assumptions allowing for non-convex and non-smooth objectives. Our proof relies on $V$-uniform ergodicity [75], which is fundamentally different from the ergodicity analysis in [1]. Under the dissipativity condition (quadratic growth of $f$), geometric ergodicity in Proposition 2.1 is not surprising; yet, it is worth highlighting that the function $f$ as well as the noise sequence require significantly less structure than what was assumed in the literature. The above step size assumption is almost standard and it is required to obtain a uniform bound on the moments of SGD iterates. We highlight that similar to the gradient descent algorithm, the step size depends on a quantity that serves as a *surrogate* condition number in our setting, namely, $L/\alpha$. Note that $\rho$ depends on $\eta$ and will typically be converging to one if $\eta \to 0$. Thus convergence in Proposition 2.1(b) can be expected to be slower when $\eta$ becomes smaller. However, smaller $\eta$ leads to a better control of the asymptotic bias (under additional regularity assumptions), see Theorems 3.1-3.3. Both of those statements (slower convergence for smaller $\eta$ but smaller bias eventually) are confirmed in our numerical experiments, see Figure 1(d,h).

Next, we state our first principal contribution, a central limit theorem for the averaged SGD iterates starting from any initial distribution, for a non-convex objective. For a test function $\phi : \mathbb{R}^d \to \mathbb{R}$, we denote the centered partial sums of $\phi$ evaluated at the SGD iterates with $S_n(\phi)$, i.e.,

$$S_n(\phi) := \sum_{k=0}^{n-1}\left[\phi\big(\theta_k^{(\eta)}\big) - \pi_\eta(\phi)\right], \quad \text{where} \quad \pi_\eta(\phi) := \int \phi(x)d\pi_\eta(x)\,.$$

**Theorem 2.1** (CLT). *Let the Assumptions 2.1-2.3 hold. For a step size $\eta$ and a test function $\phi$ satisfying the conditions in Proposition 2.1, we define $\sigma_{\pi_\eta}^2(\phi) := \lim_{n\to\infty} \frac{1}{n}\mathbb{E}_{\pi_\eta}\big[S_n^2(\phi)\big]$. Then,*

$$n^{-1/2}S_n(\phi) \xrightarrow{d} \mathcal{N}\big(0, \sigma_{\pi_\eta}^2(\phi)\big)\,.$$

The above result characterizes the fluctuations of a test function $\phi$ averaged across SGD iterates, even when the objective function is both non-convex and non-smooth. The asymptotic variance in the above CLT can be equivalently stated in another compact form. If we define the centered test function as $h(\theta) = \phi(\theta) - \pi_\eta(\phi)$, the asymptotic variance can be written as

$$\sigma_{\pi_\eta}^2(\phi) = 2\pi_\eta(h\hat{h}) - \pi_\eta(h^2), \quad \text{where} \quad \hat{h} = \sum_{k=0}^{\infty}\mathbb{E}\Big[h\big(\theta_k^{(\eta)}\big)\Big].$$

Indeed, this is the variance we compute at the end of our proof in Section A. However, the expression in Theorem 2.1 is obtained by simply applying [58, Thm 21.2.6]. For the case of strongly convex functions with decreasing step size schedule, it is well-known from the works of [28, 38] that the limiting variance of the averaged SGD iterates achieves the Cramer-Rao lower bound for parameter estimation; see also [76, 30] for non-asymptotic rates in various metrics. The question of providing lower bounds for the limiting variance of the critical points in the non-convex setting is extremely subtle, and is often handled on a case-by-case basis. We refer the interested reader to [77, 78, 12].

There are several important implications of the above CLT for constructing CIs in practice. First note that, following the standard construction in inference, one can write the distribution of the sample mean approximately as $n^{-1}S_n(\phi) \approx \mathcal{N}\big(0, n^{-1}\sigma^2_{\pi_\eta}(\phi)\big)$. Here, one needs to estimate the population quantity, the asymptotic variance $\sigma^2_{\pi_\eta}(\phi)$, for the purpose of obtaining CIs. In Section 5, we discuss three strategies for estimating this quantity, which could be eventually used for inference in practice. A theoretical analysis of the proposed approaches in Section 5 is beyond the scope of this work.

# 3 Bias of the Constant Step Size SGD

Proposition 2.1(b) shows that the expectation of a test function evaluated at the $k$-th iterate converges exponentially fast to the expected value of the stationary distribution $\pi_\eta$. Therefore, a complete characterization of the properties of the SGD requires a control over the asymptotic bias $\pi_\eta(\phi) - \phi(\theta^*)$ for a critical point $\theta^*$. It turns out that this bias behavior is intimately related to the local properties of the objective around its critical points. Therefore, under the mild assumptions that yield the CLT, one cannot expect a tight control over the bias. This section contains three types of bias analyses under different local growth conditions on the objective function $f$, characterizing the bias behavior in various non-convex and convex settings. We further note that without local regularity conditions, it is still possible to show that the SGD iterates (2) move towards a compact ball containing all critical points exponentially fast; a formal statement of this result along with a corresponding discussion is provided in Proposition B.1, which is deferred to Appendix B. Throughout this section, we make a slightly stronger assumption on the noise sequence.

**Assumption 3.1** (Fourth moment of the noise). *Gradient noise sequence $\{\xi_k\}_{k\geq 1}$ satisfies Assumption 2.3, and $\mathbb{E}\big[\|\xi_1(\theta)\|^4\big] \leq L_\xi(1 + \|\theta\|^4)$, for any $\theta \in \mathbb{R}^d$, where $L_\xi$ is as in Assumption 2.3.*

**Localized Dissipativity Condition:** We now introduce the generalized dissipativity condition which, in addition to the quadratic tail growth property enforced in Assumption 2.2, imposes a local growth within some compact region, around the unique critical point $\theta^*$.

**Assumption 3.2** (Localized dissipativity). *The objective function $f$ satisfies*

$$\langle \nabla f(\theta),\, \theta - \theta^* \rangle \geq \begin{cases} \alpha\|\theta - \theta^*\|^2 - \beta & \|\theta - \theta^*\| \geq R \\ g\big(\|\theta - \theta^*\|\big) & \|\theta - \theta^*\| < R, \end{cases}$$

*where $\theta^* \in \mathbb{R}^d$ is the unique minimizer of $f$, $R := \frac{\delta}{\alpha} + \sqrt{\frac{\beta}{\alpha}}$ with $\delta \in (0, \infty)$, $g : [0, \infty) \to [0, \infty)$ is a convex function with $g(0) = 0$ whose inverse exists.*

If $g(x) = x^2$, the objective function is *locally* strongly convex. However, the above assumption covers a wide range of objectives with different local growth rates depending on the function $g$. Next, we show that the above assumption along with the assumptions leading to the CLT is sufficient to establish an algorithmic control over the bias with a sufficiently small step size.

**Theorem 3.1.** *Let the Assumptions 2.1, 3.1, and 3.2 hold. Then SGD iterates with step size satisfying $0 < \eta < c_{L,\alpha}$ for $c_{L,\alpha}$ in (16) admit the stationary distribution $\theta^{(\eta)} \sim \pi_\eta$ which satisfies*

$$\mathbb{E}\big[\|\theta^{(\eta)} - \theta^*\|\big] \leq \tfrac{C}{\delta}\eta + g^{-1}(C\eta).$$

*Further, for a test function $\phi : \mathbb{R}^d \to \mathbb{R}$ that is $L_\phi$-Lipschitz, the bias satisfies*

$$\big|\pi_\eta(\phi) - \phi(\theta^*)\big| \leq L_\phi\big(C\eta/\delta + g^{-1}(C\eta)\big),$$

*where*

$$C := 3\big(3L^2 + 3L_\xi^{1/2}(1 + (\beta/\alpha)^2)\big)\big(1 + \int \|\theta\|^2 \pi_\eta(d\theta) + \|\theta^*\|^2\big). \tag{5}$$

If the local growth is linear, i.e. $g(x) = x$, we obtain the bias $|\pi_\eta(\phi) - \phi(\theta^*)| \leq \mathcal{O}(\eta)$. If local growth is quadratic, i.e. $g(x) = x^2$, the growth is *locally* slower than the linear case; thus, we get the bias control $|\pi_\eta(\phi) - \phi(\theta^*)| \leq \mathcal{O}(\eta^{1/2})$, which is worse in step size dependency; it reduces to the bound derived in [1, Lemma 10]. We highlight that [79] proves the following lower bound: $\liminf_{k \to \infty} \mathbb{E}[\|\theta_k^{(\eta)} - \theta^*\|^2]^{1/2} \geq c\eta^{1/2}$ for some $c > 0$ under the assumption of Lipschitz gradients. This is in line with our findings since Lipschits gradients imply $g(x) \leq x^2$ for small $x$.

**Generalized Łojasiewicz Condition:** In this section we work with a generalization of the commonly used Łojasiewicz condition in optimization.

**Assumption 3.3** (Generalized Łojasiewicz condition)**.** *The objective function $f$ has a critical point $\theta^*$ and it satisfies*

$$\|\nabla f(\theta)\|^2 \geq \begin{cases} \gamma\{f(\theta) - f(\theta^*)\} & \|\theta - \theta^*\| \geq R \\ g(f(\theta) - f(\theta^*)) & \|\theta - \theta^*\| < R, \end{cases}$$

*where $\gamma, R > 0$, and $g : [0, \infty) \to [0, \infty)$ is a convex function with $g(0) = 0$ whose inverse exists.*

In the case $g(x) = x^\kappa$ with $\kappa \in [1, 2)$, for example, the above condition is termed as the Łojasiewicz inequality [80], and for $\kappa = 1$, it reduces to the well-known Polyak-Łojasiewicz (PL) inequality [81]. Note that this inequality implies that every critical point is a global minimizer; yet, it does not imply the existence of a unique critical point.

The next result establishes an algorithmically controllable bias bound in terms of the step size.

**Theorem 3.2.** *Let the Assumptions 2.1,2.2, 3.1, and 3.3 hold, and the Hessian satisfies $\|\nabla^2 f(\theta)\| \leq \tilde{L}(1 + \|\theta\|)$, $\forall \theta \in \mathbb{R}^d$ and some $\tilde{L} > 0$. Then, the SGD iterates with a step size satisfying $0 < \eta < \frac{2}{\tilde{L}} \wedge c_{L,\alpha} \wedge c_{L,\alpha}^\dagger \wedge 1$ for $c_{L,\alpha}, c_{L,\alpha}^\dagger$ in (16) have the stationary distribution $\pi_\eta$,*

$$\pi_\eta(f) - f(\theta^*) \leq g^{-1}\left(\frac{2M\eta}{2-\tilde{L}\eta}\right) + \frac{2M\eta}{2-\tilde{L}\eta}\,,$$

*where* $M := 12\tilde{L}\left(L + L_\xi^{1/2} + L_\xi^{1/4}\right)^2\left(1 + m + m^{3/4} + \int \|\theta\|^2\pi_\eta(d\theta)\right)$ *with*

$$m := \frac{8}{7\alpha}\left[\left(\beta + 6L^2 + 3L_\xi^{1/2} + 16\right)\int \|\theta\|^2\pi_\eta(d\theta) + 16L^4 + 2L_\xi + 128L^6 + 8L_\xi^{3/2}\right].$$

*Additionally, if the test function is given as $\phi = \tilde{\phi} \circ f$ for a $L_{\tilde{\phi}}$-Lipschitz function $\tilde{\phi}$, it holds that*

$$\left|\pi_\eta(\phi) - \phi(\theta^*)\right| \leq L_{\tilde{\phi}}\left\{g^{-1}\left(\frac{2M\eta}{2-\tilde{L}\eta}\right) + \frac{2M\eta}{2-\tilde{L}\eta}\right\}.$$

For smooth objectives with Lipschitz gradient, [81] provides a linear rate under the PL-inequality (see also [82, Lemma 2]), which yields the asymptotic bias $|\pi_\eta(\phi) - \phi(\theta^*)| \leq \mathcal{O}(\eta)$. The above result recovers their findings as a special case, and provides a considerable generalization.

**Convexity:** To make the analysis of constant step size SGD complete, we digress from the main theme of this paper and consider this algorithm in the (non-strongly) convex regime, for which there is no bias characterization known to authors. We show that, under the convexity assumption, one can achieve the same bias control as in the case of PL-inequality.

**Theorem 3.3.** *Let the Assumptions 2.1,2.2, and 3.1 hold for a convex function $f$. Then, the SGD iterates with a step size $0 < \eta < c_{L,\alpha}$ for $c_{L,\alpha}$ in (16) admit the stationary distribution $\pi_\eta$, and for a minimizer $\theta^*$ it satisfies*

$$\pi_\eta(f) - f(\theta^*) \leq C\eta\,,$$

*for $C$ in (5). Further, if the test function is given as $\phi = \tilde{\phi} \circ f$ for a $L_{\tilde{\phi}}$-Lipschitz function $\tilde{\phi}$, then,*

$$\left|\pi_\eta(\phi) - \phi(\theta^*)\right| \leq L_{\tilde{\phi}}C\eta\,.$$

Convexity implies that any critical point $\theta^*$ is a global minimizer, which is similar to the PL-inequality; yet, it does not imply a unique minimizer unlike strong convexity. The resulting step size dependency of the bias is the same as in the case of PL-inequality, which is because both of these conditions assert a similar gradient-based domination criterion on the sub-optimality. That is, we have in the convex case $\langle\nabla f(\theta), \theta - \theta^*\rangle \geq f(\theta) - f(\theta^*)$, and in the case of PL-inequality $\gamma^{-1}\|f(\theta)\|^2 \geq f(\theta) - f(\theta^*)$.

# 4  Examples and Numerical Studies

We now demonstrate the asymptotic normality and bias in non-convex optimization with two examples arising in robust statistics for which our assumptions can be verified. We consider the online SGD setting with the update rule: $\theta_{k+1}^{(\eta)} = \theta_k^{(\eta)} - \frac{\eta}{b_k} \sum_{j=1}^{b_k} \nabla F(\theta_k^{(\eta)}, Z_j)$, for $k \geq 0$, with independent samples $Z_j \sim P(Z)$ used to estimate the true gradient in each iteration $k$ ; and also the semi-stochastic setting, where the noise sequence $\{\xi_k(\theta)\}_{k \geq 1}$ is independent of $\theta$ and is simply a sequence of i.i.d. random vectors – such a setting helps verifying our assumptions more explicitly.

## 4.1  Regularized MLE for heavy-tailed linear regression

While the least-squares loss function is common in the context of linear regression, it is well-documented that it suffers from robustness issues when the error distribution of the model is heavy-tailed [83]. Indeed in fields like finance, oftentimes the Student's $t$-distribution is used to model the heavy-tailed error [84]. In this case, defining the random vector $Z := (X, Y)$, the stochastic optimization problem in (3) is given by the expectation of the function $F(Z, \theta) := \log\big(1 + (Y - \langle X, \theta \rangle)^2\big) + \frac{\lambda}{2}\|\theta\|^2$, which is non-convex (as a function of $\theta$) for small penalty levels $\lambda > 0$. Correspondingly, given $n$ independent and identically distributed samples $(\mathbf{x}_i, y_i)$, the finite-sum version of the optimization problem corresponds to minimizing the following objective function

$$f(\theta) := \frac{1}{2m} \sum_{i=1}^m \log\big(1 + (y_i - \langle \mathbf{x}_i, \theta \rangle)^2\big) + \frac{\lambda}{2}\|\theta\|^2. \tag{6}$$

We consider the finite-sum setup and we verify our assumptions and empirically demonstrate our results on CLT as well as the bias in a clean manner.

### 4.1.1  Semi-stochastic Gradient Descent

In the experiments, $\mathbf{X} := (\mathbf{x}_1, \ldots, \mathbf{x}_m)^\top \in \mathbb{R}^{m \times d}$ represents a fixed design matrix generated from $\mathbf{X}_{ij} \sim \text{Bernoulli}(\pm 1)/\sqrt{d}$, and $\mathbf{y} := (y_1, \ldots, y_m)^\top \in \mathbb{R}^m$ represents the response vector generated according to the linear model $y_i = \langle \mathbf{x}_i, \theta_{\text{true}} \rangle + \varepsilon$ with $(\theta_{\text{true}})_i \overset{\text{iid}}{\sim} \text{Unif}(0, 1)$, and $\varepsilon$ is Student-t distributed (df $= 10$) noise. We choose $m = 5000$, $d = 10$, and the Lipschitz test function $\phi(\theta) = \|\theta\|$ unless stated otherwise.

**Asymptotic normality:** Fig. 1-(a,b,c,d) demonstrates the normality and the bias of SGD with heavy-tailed gradient noise distributed as Student-t (df $= 5$). Each plot has two density curves where red and blue curves in Fig. 1-(a,b) respectively correspond to initializations with $\theta_0 = (1, \ldots, 1)^\top$ and $\theta_0' = (1.5, \ldots, 1.5)^\top$ with step size $\eta = 0.3$; green and orange curves in Fig. 1-c correspond to step sizes $\eta = 0.2$ and $\eta' = 0.3$ with initialization $\theta_0$. All experiments are based on 4000 Monte Carlo runs. We observe in Fig. 1-a that different initializations have an early impact on the normality when the number of iterations is moderate. However, when SGD is run for a longer time, this effect is removed as in Fig. 1-b. Lastly, Fig.1-c demonstrates the effect of step size on the normality, where the means are different for different step sizes as they depend on the stationary distribution $\pi_\eta$. Indeed, the above results are not surprising as one can verify that the assumptions of Theorem 2.1 are satisfied.

**Lemma 4.1.** *The objective function (6) satisfies Assumptions 2.1 and 2.2. Further, Assumption 2.3 is also satisfied with the Student-t distributed (df $= 10$) noise.*

**Bias:** In order to demonstrate the bias behavior without speculation, one needs the global minimum $\theta^*$ of the non-convex problem. Therefore, we simplify the problem (6) to another non-convex problem

$$f(\theta) := \frac{1}{2} \log\big(1 + \|\theta\|^2\big) + \frac{\lambda}{2}\|\theta\|^2 . \tag{7}$$

Notice that the general structure is the same, with no data, and $\theta^*$ is known, i.e. $\theta^* = 0$. We choose the test function $\phi(\theta) = \tilde{\phi} \circ f(\theta)$, where $\tilde{\phi}(x) = 1/(1 + e^{-x})$ is Lipschitz. Fig. 1-(d) demonstrates how the bias $\pi_\eta(\phi) - \phi(\theta^*)$ changes over iterations, where different curves correspond to different step sizes. We notice that larger step size provides fast initial decrease; yet the resulting asymptotic bias is larger which aligns with our theory – indeed, a smaller asymptotic bias for a smaller step size $\eta$ is predicted by Theorem 3.2 while slower convergence can be expected given the discussion after Proposition 2.1. The following lemma proves that our assumptions are satisfied for this objective.

**Lemma 4.2.** *The objective function (7) is non-convex when $\lambda$ is sufficiently small, and it satisfies Assumptions 2.1, 2.2, and 3.3. Further, Assumption 3.1 is also satisfied for this example.*

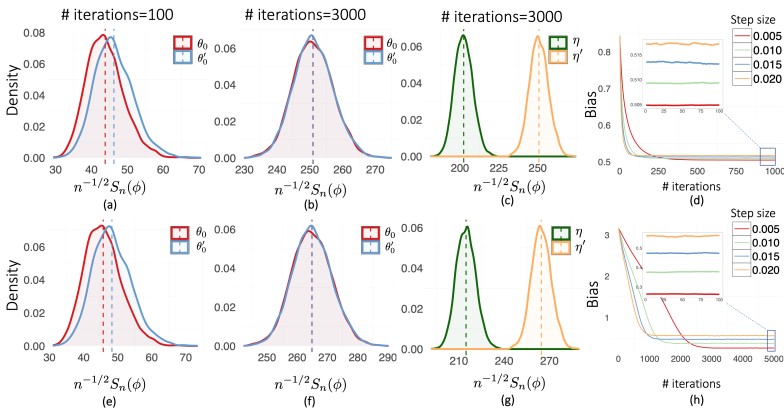

Figure 1: First and second rows correspond to non-convex examples in Sections 4.1.1 and 4.2.1, respectively. Figures (a,b), (e,f) show the density of $n^{-1/2}S_n(\phi) = n^{-1/2}\sum_{k=1}^{n}\phi(\theta_k^{(\eta)})$ with different initializations (red, blue) for different number of iterations. Figures (c,g) show the same density with different step sizes. Figures (d,h) show the evolution of bias against iterations.

### 4.1.2 Online Stochastic Gradient Descent

For our online SGD experiments, we use $b_k = 2$, for all $k$ to obtain the stochastic gradient. We also experimented with $m_k = 1, 10, 50$ and observed similar behavior. The distribution of the random vector $Z = (X, Y) \in \mathbb{R}^{d+1}$, is as follows: Each coordinate of the vector $X \in \mathbb{R}^d$, is generated as Bernoulli$(\pm 1)/\sqrt{d}$ and given vector $X$, the response $Y \in \mathbb{R}$ is generated according to the linear model $Y = \langle X, \theta_{\text{true}} \rangle + \varepsilon$ with each coordinate of $\theta_{\text{true}} \in \mathbb{R}^d$ generated from $\text{Unif}(0, 1)$, and fixed, and $\varepsilon \in \mathbb{R}$ is Student-t (df = 10) noise. We choose $d = 10$, and set a burn-in period of size 100.

**Asymptotic normality:** Fig. 2-(a,b,c) demonstrates the normality of online SGD. Each plot has two density curves where red and blue curves in Fig. 2-(a,b) respectively correspond to initializations with $\theta_0 = (1, \ldots, 1)^\top$ and $\theta_0' = (2.5, \ldots, 2.5)^\top$ with step size $\eta = 0.3$; green and orange curves in Fig. 2-c correspond to step sizes $\eta = 0.2$ and $\eta' = 0.3$ with initialization $\theta_0$. All experiments are based on 4000 Monte Carlo runs. We observe in Fig. 2-a that different initializations have an early impact on the normality when the number of iterations is moderate. However, when SGD is run for a longer time, this dependence is removed as in Fig. 2-b. Lastly, Fig.2-c demonstrates the effect of step size on the normality, where the means are different for different step sizes as they depend on the stationary distribution $\pi_\eta$. Indeed, all these observations are as predicted by our theory.

### 4.2 Regularized Blake-Zisserman MLE for corrupted linear regression

While the above example was based on linear-regression with heavy-tailed noise, we now consider the case of heavy-tailed regression with corrupted noise. In this setup, the noise model in linear regression is assumed to be Gaussian, but a fraction of the noise vectors are assumed to be corrupted in the sense that they are drawn from a uniform distribution. Such a scenario arises in visual reconstruction problems; see for example [85, 33] for details. In this case, defining the random vector $Z := (X, Y)$, the stochastic optimization problem in (3) is given by the expectation of the function $F(Z, \theta) := \log\left(\nu + e^{-(Y - \langle X, \theta \rangle)^2}\right) + \frac{\lambda}{2}\|\theta\|^2$, for $\nu > 0$. Similar the previous case, we also consider the finite-sum version: Given $n$ independent and identically distributed samples $(\mathbf{x}_i, y_i)$, it corresponds to minimizing the following objective function

$$f(\theta) = -\frac{1}{2m}\sum_{i=1}^{m}\log\left(\nu + e^{-(y_i - \langle \mathbf{x}_i, \theta \rangle)^2}\right) + \frac{\lambda}{2}\|\theta\|^2, \quad \nu > 0. \tag{8}$$

### 4.2.1 Semi-stochastic Gradient Descent

**Asymptotic normality:** In the experiments, we use the same setup and parameters as in Section 4.1.1. Fig 1-(e,f,g) demonstrates the asymptotic normality of the SGD with heavy-tailed gradient noise Student-t(df = 6). The experimental setup is the same as the previous example with the same values for $\theta_0, \theta_0', \eta, \eta'$. We observe the early impact of initialization in Fig 1-a, the clear normality in Fig. 1-b, and the effect of step size on CLT in Fig.1-c. These observations also align with our theory since this objective also satisfies our assumptions.

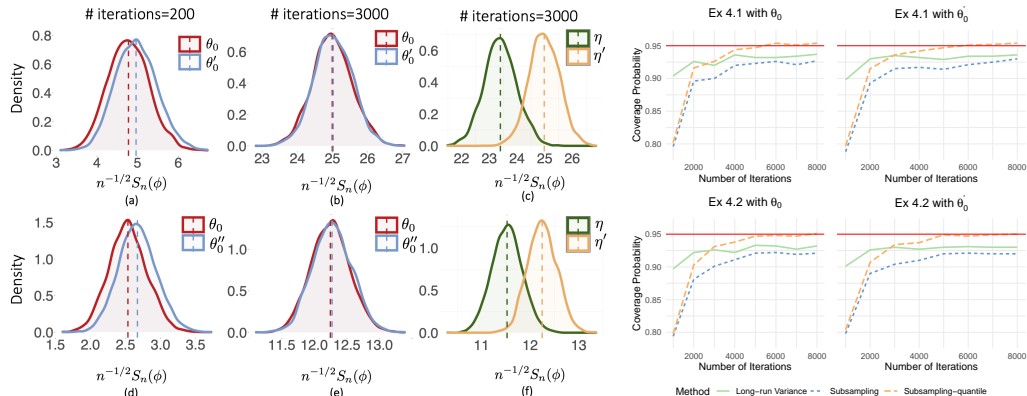

Figure 2: **Left:** First and second rows correspond to non-convex examples in Sections 4.1.2 and 4.2.2, respectively. Figures (a,b), (d,e) show the density of $n^{-1/2}S_n(\phi) = n^{-1/2}\sum_{k=1}^{n}\phi(\theta_k^{(\eta)})$ with different initializations (red, blue) for different number of iterations. Figures (c,f) show the same density with different step sizes. **Right:** Coverage probabilities for Subsampling quantile, Subsampling var, and Long-run var as functions of the number of iterations. Subsampling quantile method outmatches the others in terms of coverage probability and achieves the nominal level with larger iterations.

**Lemma 4.3.** *The objective function* (8) *satisfies Assumptions 2.1, 2.2. Further, Assumption 2.3 is also satisfied with the Student-t (df = 10) noise.*

**Bias:** Similar to the previous example, we simplify the problem so that we can compute the bias $\pi_\eta(\phi) - \phi(\theta^*)$. We consider the function

$$f(\theta) := -\frac{1}{2}\log\left(\nu + e^{-\|\theta\|^2}\right) + \frac{\lambda}{2}\|\theta\|^2, \quad \nu > 0. \tag{9}$$

We observe in Fig.1-h that smaller step sizes lead to smaller asymptotic bias. One can verify that this can be predicted from Theorem 3.1.

**Lemma 4.4.** *The objective function* (9) *is non-convex when $\lambda$ is sufficiently small, and it satisfies Assumptions 2.1 and 3.2. Further, Assumption 3.1 is also satisfied for this example.*

### 4.2.2   Online Stochastic Gradient Descent

**Asymptotic normality:** In the experiments, we use the same setup as in Section 4.1.2. Fig. 2-(d,e,f) demonstrates the normality of online SGD. Each plot has two density curves where red and blue curves in Fig. 2-(d,e) respectively correspond to initializations with $\theta_0 = (1, \ldots, 1)$ and $\theta_0'' = (1.5, \ldots, 1.5)$ with step size $\eta = 0.3$; green and orange curves in Fig. 2-c correspond to step sizes $\eta = 0.2$ and $\eta' = 0.3$ with initialization $\theta_0$. All experiments are based on 4000 Monte Carlo runs. We observe in Fig. 2-d that different initializations have an early impact on the normality when the number of iterations are moderate. However, when SGD is run for a longer time, this effect is removed as in Fig. 2-e. Lastly, Fig.2-f demonstrates the effect of step size on the normality, where the means are different for different step sizes as they depend on the stationary distribution $\pi_\eta$.

## 5   Discussions

By leveraging the connection between constant step size SGD and Markov chains [1], we provided theoretical results characterizing the fluctuations and bias of SGD for non-convex and non-smooth optimization which arises frequently in statistical learning.

**Estimating the Asymptotic Variance:** As discussed in Section 2, in order to use the established CLT to compute CIs in practice, the population expectation $\pi_\eta(\phi)$ and asymptotic variance $\sigma^2_{\pi_\eta}(\phi)$ have to be estimated. We suggest the following three ways to do so:

- Estimate them based on sample average of a single trajectory of SGD iterates, i.e., the mean $\pi_\eta(\phi)$ is estimated as $n^{-1}\sum_{k=0}^{n-1}\phi(\theta_k^{(\eta)})$, and the asymptotic variance $\sigma^2_{\pi_\eta}(\phi)$ can be estimated by adopting the online approach of [86] to the constant step size setting. The variance $\sigma^2_{\pi_\eta}(\phi)$ can

also be estimated by the Newey-West long-run variance estimation [87, 88] or empirical variance estimation based on sub-sampling [89, Sections 4.2 and 4.6] for a single trajectory.

- First run $N$ parallel SGD trajectories and compute the average of each trajectory, to obtain $N$ independent observations from the stationary distribution $\pi_\eta$. Next, use the $N$ observations to compute the sample mean and the sample variance estimators for $\pi_\eta(\phi)$ and $\sigma^2_{\pi_\eta}(\phi)$.
- Leverage the online bootstrap and variance estimation approaches proposed in [43, 41, 90] for the constant step size SGD setting in order to obtain estimates for $\pi_\eta(\phi)$ and $\sigma^2_{\pi_\eta}(\phi)$.

As a confirmation of the practicability of constructing CIs, we provide preliminary experimental results for constructing CIs with minibatch SGD. We consider the data generation setup described in Sections 4.1 and 4.2 with step size 0.3, and run online SGD with batch size 2. In each run, the first 200 values are discarded. CIs are constructed for each trajectory based on sub-sampling with empirical CDF (Subsampling quantile) and variance estimation (Subsampling var) [89, Sections 4.2 and 4.6], and Newey-West long-run variance estimation (Long-run var) with data-driven bandwidth selection [87, 88]. Empirical coverage results as a function of iteration numbers (nominal level = 95%, 4000 Monte Carlo replications) for the three methods and different initializations ($\theta_0 = (1, \ldots, 1)^\top$ and $\theta'_0 = (1.5, \ldots, 1.5)^\top$) are reported in Figure 2 (right). A non-asymptotic justification of the relative merits of the above variance estimation approaches are left as future work.

## Acknowledgements

MAE is partially funded by NSERC Grant [2019-06167], Connaught New Researcher Award, CIFAR AI Chairs program, and CIFAR AI Catalyst grant. KB is partially supported by a seed grant from the Center for Data Science and Artificial Intelligence Research, UC Davis. SV is partially supported by a discovery grant from NSERC of Canada and a Connaught New Researcher Award. The authors thank Yichen Zhang for helpful comments on an earlier version of this manuscript.

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
