# An Analysis of Constant Step Size SGD in the Non-convex Regime: Asymptotic Normality and Bias

**SUPPLEMENTARY DOCUMENT**

## A Proofs for Sections 2 and 3

### A.1 Preliminaries and Additional Notations

Note that the sequence of iterates $\{\theta_k^{(\eta)}\}_{k\geq 0}$ is a homogeneous Markov chain [1]. We denote the (sub-)$\sigma$-algebra (of $\mathcal{F}$) of events up to and including the $k$-th iteration as $\mathcal{F}_k$. By definition, the discrete-time stochastic process defined in (2) is adapted to its natural filtration $\{\mathcal{F}_k\}_{k\geq 0}$. We denote the Markov kernel on $(\mathbb{R}^d, \mathcal{B}(\mathbb{R}^d))$ associated with SGD iterates (2) by $P$ with

$$P(\theta_k^{(\eta)}, A) = \mathbb{P}(\theta_{k+1}^{(\eta)} \in A | \theta_k^{(\eta)}) \ \ \mathbb{P} - a.s., \quad \forall A \in \mathcal{B}(\mathbb{R}^d), k \geq 0 \, .$$

Define the $k$-th power of this kernel iteratively: define $P^1 := P$, and for $k \geq 1$, for all $\tilde{\theta} \in \mathbb{R}^d$ and $A \in \mathcal{B}(\mathbb{R}^d)$, define

$$P^{k+1}(\tilde{\theta}, A) := \int_{\mathbb{R}^d} P(\tilde{\theta}, d\theta) P^k(\theta, A) \, .$$

For any function $\phi : \mathbb{R}^d \to \mathbb{R}$ and $k \geq 0$, define the measurable function $P^k\phi(\theta) : \mathbb{R}^d \to \mathbb{R}$ for all $\theta \in \mathbb{R}^d$ via

$$P^k\phi(\theta) = \int \phi(\tilde{\theta}) P^k(\theta, d\tilde{\theta}) \, .$$

Given the $L_\phi$-Lipschitz function $\phi : \mathbb{R}^d \to \mathbb{R}$ and the expectation of $\phi$ under the stationary measure $\pi_\eta$, define the function $h$ as

$$\begin{aligned} h : \mathbb{R}^d &\to \mathbb{R} \\ \theta &\mapsto \phi(\theta) - \pi_\eta(\phi) \, . \end{aligned}$$

Note that $\pi_\eta(h) = 0$ and $h$ is $L_\phi$-Lipschitz. Define the partial sum $S_n(\phi) := \sum_{k=0}^{n-1} h(\theta_k^{(\eta)})$. Moreover, we define

$$\bar{\theta}_\eta := \int_{\mathbb{R}^d} \theta \, d\pi_\eta(\theta) \, .$$

### A.2 Proofs of Proposition 2.1 and Theorem 2.1

We start with some preliminary results required to prove the CLT.

**Lemma A.1.** *Under Assumptions 2.1-2.3, it holds for any $\eta \in \left(0, \frac{\alpha - \sqrt{(\alpha^2 - (3L^2 + L_\xi)) \vee 0}}{3L^2 + L_\xi}\right)$ and any fixed initial point $\theta_0^{(\eta)} = \theta_0 \in \mathbb{R}^d$ that*

$$\mathbb{E}[\, \|\theta_{k+1}^{(\eta)}\|^2 + 1 | \mathcal{F}_k] \leq \alpha_\dagger (\, \|\theta_k^{(\eta)}\|^2 + 1) + \beta_\dagger \, .$$

*Here, $\alpha_\dagger \in (0,1)$ and $\beta_\dagger \in (0,\infty)$ are constants depending on $\eta$. The explicit formulas of $\alpha_\dagger, \beta_\dagger$ are given in the proof.*

*Proof of Lemma A.1.* Define $U_\eta := \frac{\alpha - \sqrt{\max\{\alpha^2 - (3L^2 + L_\xi), 0\}}}{3L^2 + L_\xi}$. Given $\eta \in (0, U_\eta)$, define

$$\alpha_\dagger = 1 + \eta^2 (3L^2 + L_\xi) - 2\eta\alpha \, ,$$

and note that with this definition $\alpha_\dagger \in (0,1)$ whenever $\eta \in (0, U_\eta)$. Then, with $\eta, \alpha_\dagger$, and the fixed initial point $\theta_0^{(\eta)} = \theta_0 \in \mathbb{R}^d$, we set

$$\beta_\dagger := \kappa(\alpha_\dagger^{1/2} - \alpha_\dagger) \, ,$$

where

$$\kappa := \frac{4\eta(\alpha+\beta)+12\eta^2 L^2}{\alpha_\dagger^{1/2}-\alpha_\dagger} \bigvee 1 \,.$$

It follows that $\beta_\dagger > 0$. Note that

$$\mathbb{E}[1+\|\theta_{k+1}^{(\eta)}\|^2|\mathcal{F}_k]$$
$$=\mathbb{E}[1+\|\theta_k^{(\eta)}-\eta\big(\nabla f(\theta_k^{(\eta)})+\xi_{k+1}(\theta_k^{(\eta)})\big)\|^2|\mathcal{F}_k]$$
$$=1+\mathbb{E}\big[\|\theta_k^{(\eta)}\|^2+\eta^2\|\nabla f(\theta_k^{(\eta)})\|^2+\eta^2\|\xi_{k+1}(\theta_k^{(\eta)})\|^2-2\eta\langle\theta_k^{(\eta)},\nabla f(\theta_k^{(\eta)})\rangle|\mathcal{F}_k\big]\,.$$

The last step follows from the Assumption 2.3. By Assumption 2.1, we have

$$\|\nabla f(\theta_k^{(\eta)})\|^2 \le L^2(1+\|\theta_k^{(\eta)}\|)^2\,.$$

Squaring both sides and using the fact that $(1+\|\theta_k^{(\eta)}\|)^2 \le 3(\|\theta_k^{(\eta)}\|^2+3)$ gives

$$\|\nabla f(\theta_k^{(\eta)})\|^2 \le 3L^2(\|\theta_k^{(\eta)}\|^2+3)\,.$$

By Assumption 2.2, we obtain

$$\langle\theta_k^{(\eta)},\nabla f(\theta_k^{(\eta)})\rangle \ge \alpha\|\theta_k^{(\eta)}\|^2-\beta\,.$$

By Assumption 2.3, it holds that

$$\mathbb{E}[\|\xi_{k+1}(\theta_k^{(\eta)})\|^2|\mathcal{F}_k] \le L_\xi(1+\|\theta_k^{(\eta)}\|^2)\,.$$

Plugging the previous three inequalities into the first display provides us with

$$\mathbb{E}[1+\|\theta_{k+1}^{(\eta)}\|^2|\mathcal{F}_k] \le 1+9\eta^2 L^2+\eta^2 L_\xi+2\eta\beta+(1+3\eta^2 L^2+\eta^2 L_\xi-2\eta\alpha)\|\theta_k^{(\eta)}\|^2\,. \quad (10)$$

Recall that $\alpha_\dagger = 1+\eta^2(3L^2+L_\xi)-2\eta\alpha$. Plugging $\alpha_\dagger$ back into the previous display yields

$$\mathbb{E}[\|\theta_{k+1}^{(\eta)}\|^2+1|\mathcal{F}_k] \le \alpha_\dagger(\|\theta_k^{(\eta)}\|^2+1)+2\eta(\alpha+\beta)+6\eta^2 L^2\,.$$

Note that $\beta_\dagger = \kappa(\alpha_\dagger^{1/2}-\alpha_\dagger)$, where

$$\kappa \ge \frac{4\eta(\alpha+\beta)+12\eta^2 L^2}{\alpha_\dagger^{1/2}-\alpha_\dagger}\,.$$

It then follows that $\mathbb{E}[\|\theta_{k+1}^{(\eta)}\|^2+1|\mathcal{F}_k] \le \alpha_\dagger(\|\theta_k^{(\eta)}\|^2+1)+\beta_\dagger$ as desired. $\qquad\square$

**Corollary A.1** (Bounded second moment). *Under the assumptions stated in Lemma A.1, with the constant step size $\eta \in \left(0,\frac{\alpha-\sqrt{(\alpha^2-(3L^2+L_\xi))\vee 0}}{3L^2+L_\xi}\right)$ the stationary distribution $\pi_\eta$ satisfies*

$$\mu_{2,\eta} := \int \|\theta\|^2\pi_\eta(d\theta) \le 3+\frac{2\beta}{\alpha}\,.$$

*Proof of Corollary A.1.* Consider the chain $\{\theta_k^{(\eta)}\}_{k\ge 0}$ starting from the stationary distribution $\pi_\eta$. By display (10), it holds that

$$\mathbb{E}[\|\theta_{k+1}^{(\eta)}\|^2] \le 9\eta^2 L^2+\eta^2 L_\xi+2\eta\beta+(1+3\eta^2 L^2+\eta^2 L_\xi-2\eta\alpha)\|\theta_k^{(\eta)}\|^2\,.$$

Using the fact that by stationarity $\mathbb{E}[\|\theta_{k+1}^{(\eta)}\|^2] = \mathbb{E}[\|\theta_k^{(\eta)}\|^2]$ and rearranging the previous display gives

$$\mathbb{E}[\|\theta_k^{(\eta)}\|^2] \le \frac{9\eta L^2+\eta L_\xi+2\beta}{2\alpha-\eta(3L^2+L_\xi)} \le 3+\frac{2\beta}{\alpha}\,.$$

$\square$

**Corollary A.2** (Lyapunov condition). *Under the assumptions stated in Lemma A.1, given the step size specified in Lemma A.1, it holds that*

$$\mathbb{E}[V(\theta_{k+1}^{(\eta)})|\mathcal{F}_k] \le \alpha_\dagger V(\theta_k^{(\eta)}) + \beta_\dagger \,,$$

*where the Lyapunov function $V(\theta)$ is defined via*

$$V(\theta) := \|\theta\|^2 + 1 \,. \tag{11}$$

*Observe that by the proof of Lemma 15.2.8 in [75] this also implies that the drift condition (V4) in [75] holds with $V$ defined above, $b = \beta_\dagger, \beta = (1 - \alpha_\dagger)/2$ and the following set $\mathcal{C}$*

$$\mathcal{C} := \left\{\theta \in \mathbb{R}^d : V(\theta) \le \frac{2\beta_\dagger}{\gamma - \alpha_\dagger}\right\}, \tag{12}$$

*for an arbitrary but fixed $\gamma \in (\alpha_\dagger^{1/2}, 1)$.*

**Corollary A.3** (Minorization condition). *Under Assumptions 2.1-2.3, given the step size specified in Lemma A.1, there exists a constant $\zeta > 0$, and a probability measure $\nu^\dagger$ (depending on $\eta$ which is suppressed in the notation) with $\nu^\dagger(\mathcal{C}) = 1$ and $\nu^\dagger(\mathcal{C}^c) = 0$, such that*

$$P(\theta, A) \ge \zeta\nu^\dagger(A)$$

*holds for any $A \in \mathcal{B}(\mathbb{R}^d)$ and $\theta \in \mathcal{C}$ for the set $\mathcal{C}$ defined in (12).*

*Proof of Corollary A.3.* Recall the definition of the markov chain (2), we have

$$\xi_{k+1}(\theta_k^{(\eta)}) = \frac{\theta_k^{(\eta)} - \theta_{k+1}^{(\eta)}}{\eta} - \nabla f(\theta_k^{(\eta)}) \,.$$

Recall that the distribution of $\xi_1(\theta)$ can be decomposed as $\mu_{1,\theta} + \mu_{2,\theta}$ where $\mu_{1,\theta}$ has density $p_\theta$. It then holds for any $\theta \in \mathbb{R}^d$ that

$$P(\theta, \mathcal{C}) = \mathbb{P}(\theta_{k+1}^{(\eta)} \in \mathcal{C}|\theta_k^{(\eta)} = \theta) \ge \int_{t \in \mathcal{C}} \frac{1}{\eta^d} p_\theta\left(\frac{\theta - t}{\eta} - \nabla f(\theta)\right) dt > 0 \,. \tag{13}$$

This implies every state in the state space is within reach of any other state over the set $\mathcal{C}$. Define the probability measure $\nu^\dagger$ with density

$$p_{\nu^\dagger}(t) := I\{\theta \in \mathcal{C}\} \frac{\inf_{\theta \in \mathcal{C}} p(t|\theta)}{\int_{t \in \mathcal{C}} \inf_{\theta \in \mathcal{C}} p(t|\theta) dt} \,,$$

and set the constant $\zeta := \int_{t \in \mathcal{C}} \inf_{\theta \in \mathcal{C}} p(t|\theta) dt$. By Assumption 2.3 and the display (13), it holds that $\zeta > 0, \nu^\dagger(\mathcal{C}) = 1$ and $\nu^\dagger(\mathcal{C}^c) = 0$. Moreover, it holds that any $A \in \mathcal{B}(\mathbb{R}^d)$ and $\theta \in \mathcal{C}$ that

$$P(\theta, A) \ge \zeta\nu^\dagger(A) \,.$$

This implies the minorization condition is met for all choices of $\eta$ given by Lemma A.1. $\square$

**Lemma A.2.** *Under Assumptions 2.1-2.3, the chain $\{\theta_k^{(\eta)}\}_{k \ge 0}$ is an aperiodic, $\psi$-irreducible, and Harris recurrent chain, with an invariant measure $\pi_\eta$.*

**Remark A.1.** *This lemma implies the chain $\{\theta_k^{(\eta)}\}_{k \ge 0}$ is positive.*

*Proof of Lemma A.2.* **Step 1**: We show that the chain $\{\theta_k^{(\eta)}\}_{k \ge 0}$ is aperiodic. By Assumption 2.3, there does not exist $d \ge 2$ and a partition of size $d + 1$ such that $\mathcal{B}(\mathbb{R}^d) = (\dot{\cup}_{i=1}^d D_i)\dot{\cup}N$, where $\dot{\cup}$ denotes the disjoint union, and $N$ is a $\psi$-null (transient) set, such that $P(\theta, D_{i+1}) = 1$ holds for $\psi$-a.e. $\theta \in D_i$. Thus, the largest *period* of the chain defined in (2) is 1, which implies the chain is aperiodic.

**Step 2**: We show that the chain $\{\theta_k^{(\eta)}\}_{k \ge 0}$ is $\psi$-irreducible, and recurrent with an invariant probability measure. We note that by Assumption 2.3, there exists some non-zero $\sigma$-finite measure $\psi$ on $(\mathbb{R}^d, \mathcal{B}(\mathbb{R}^d))$ such that for any $\theta \in \mathbb{R}^d$ and any $A \in \mathcal{B}(\mathbb{R}^d)$ with $\psi(A) > 0$, it holds that

$$\mathbb{P}(\theta_{k+1}^{(\eta)} \in A|\theta_k^{(\eta)} = \theta) \ge \int_{\tilde{\theta} \in A} \frac{1}{\eta^d} p_\theta\left(\frac{\theta - \tilde{\theta}}{\eta} - \nabla f(\theta)\right) d\tilde{\theta} > 0 \,,$$

where $p_\theta$ was defined in Assumption 2.3. This implies the Markov chain defined in (2) is $\psi$-irreducible. By the Lyapunov condition established in Corollary A.2, part (iii) of Theorem 15.0.1 in [75] holds. It then follows by condition (i) of this theorem that the chain $\{\theta_k^{(\eta)}\}_{k \geq 0}$ is recurrent with an invariant probability measure $\pi_\eta$.

**Step 3**: We show that the chain is Harris recurrent. Define the hitting time $\tau_\mathcal{C} := \inf\{n > 0 : \theta_n^{(\eta)} \in \mathcal{C}\}$, where the set $\mathcal{C}$ is defined in (12). By Corollary A.4 in [72], it holds for any fixed $\theta_0^{(\eta)} = \theta_0 \in \mathbb{R}^d$ that

$$\mathbb{P}(\tau_\mathcal{C} < \infty) = 1 \,.$$

By Proposition 10.2.4 in [58], the chain is Harris recurrent. $\qquad \square$

Now, we are ready to prove Proposition 2.1.

*Proof of Proposition 2.1.* (a) By Lemma A.2, the chain $\{\theta_k^{(\eta)}\}_{k \geq 0}$ is an aperiodic Harris recurrent chain, with an invariant measure $\pi_\eta$. Note that the chain is also positive. Thus condition (i) of Theorem 13.0.1 in [75] is satisfied and this implies the existence of a unique invariant measure $\pi_\eta$. The fact that this stationary distribution has a finite second moment was established in Corollary A.1.

(b) By Lemma A.2, the iterates $\{\theta_k^{(\eta)}\}_{k \geq 0}$ are realiztions from a $\psi$-irreducible and aperiodic chain. Note that

$$\begin{aligned} |\phi(\theta)| &\leq \kappa_\phi(1 + \|\theta\|) \\ &\leq 2\kappa_\phi\sqrt{1 + \|\theta\|^2} \\ &\leq 2\kappa_\phi V(\theta) \,. \end{aligned}$$

By Corollary A.2, the condition (iv) of Theorem 16.0.1 in [75] with $V(\theta) = 2\kappa_\phi(1 + \|\theta\|^2)$ is fulfilled. By part (ii) in that theorem, it holds that for fixed $\theta_0^{(\eta)} = \theta_0 \in \mathbb{R}^d$

$$|P^k\phi(\theta_0) - \pi_\eta(\phi)| \leq \kappa\rho^k V(\theta_0) \,,$$

where $\rho \in (0, 1), \kappa > 0$ are constants depending on $\phi$. $\qquad \square$

We now prove Theorem 2.1. In order to do so, we first derive the central limit theorem for the function $h$ when the Markov chain starting from its stationary distribution $\pi_\eta$.

**Lemma A.3** (CLT with stationary initial distribution). *Assume Assumptions 2.1-2.3 hold. For any step size* $\eta \in \left(0, \frac{\alpha - \sqrt{(\alpha^2 - (3L^2 + L_\xi)) \vee 0}}{3L^2 + L_\xi}\right)$, *it holds that*

$$n^{-1/2} S_n(\phi) \xrightarrow[\mathbb{P}_{\pi_\eta}]{} \mathcal{N}(0, \sigma_{\pi_\eta}^2(\phi)) \,,$$

*where* $\sigma_{\pi_\eta}^2(\phi) = 2\pi_\eta(h\hat{h}) - \pi_\eta(h^2)$ *with* $\hat{h} = \sum_{k=0}^\infty P^k h$.

*Proof of Lemma A.3.* We prove the claim by appealing to Theorem 17.0.1 in [75]. In order to do so, we first show that the chain $\{\theta_k^{(\eta)}\}_{k \geq 0}$ is $V$-uniformly ergodic, where the function $V$ is defined in (11). Then, we establish the CLT by employing Theorem 17.0.1 in [75].

**Step 1**: We show that the chain $\{\theta_k^{(\eta)}\}_{k \geq 0}$ is $V$-uniformly ergodic. By Lemma A.2 and Proposition 2.1, the chain $\{\theta_k^{(\eta)}\}_{k \geq 0}$ is positive Harris recurrent with a unique stationary distribution $\pi_\eta$. Note that the chain $\{\theta_k^{(\eta)}\}_{k \geq 0}$ is also $\psi$-irreducible and aperiodic. By Corollary A.2, condition (iv) of Theorem 16.0.1 in [75] is satisfied. Then, it follows from part (i) of this theorem that the iterates $\{\theta_k^{(\eta)}\}_{k \geq 0}$ is $V$-uniformly ergodic.

**Step 2**: We now establish the CLT for the averaged SGD iterates starting from the stationary distribution $\pi_\eta$. Note that for the test function $\phi(\theta)$, it holds for any $\theta \in \mathbb{R}^d$ that

$$|\phi(\theta)| \leq \kappa_\phi(1 + \|\theta\|) \leq 2\kappa_\phi\sqrt{1 + \|\theta\|^2} \,,$$

which implies

$$|\phi(\theta)|^2 \leq 4\kappa_\phi^2 V(\theta) \,.$$

Thus the conditions required to leverage Theorem 17.0.1 (ii), (iv) with $g(\theta) = \phi(\theta)$ in [75] are satisfied. Hence, by Theorem 17.0.1 in [75], we obtain

$$\frac{1}{\sqrt{n}} \sum_{k=0}^{n-1} h(\theta_k^{(\eta)}) \xrightarrow[\mathbb{P}_{\pi_\eta}]{} \mathcal{N}(0, \sigma_{\pi_\eta}^2(\phi)),$$

where $\sigma_{\pi_\eta}^2(\phi) = 2\pi_\eta(h\hat{h}) - \pi_\eta(h^2) > 0$. $\qquad \square$

*Proof of Theorem 2.1.* By Lemma A.2 and Lemma A.3, the desired result follows readily from Proposition 17.1.6 in [75]. $\qquad \square$

## A.3   Proofs of Proposition B.1, Theorem 3.1, Theorem 3.2, and Theorem 3.3

We need the following auxiliary lemmas.

**Lemma A.4.** *Assumptions 2.1 and 2.2 implies*

$$\langle \nabla f(\theta), \theta - \theta^* \rangle \geq \alpha' \|\theta - \theta^*\|^2 - \beta',$$

*where $\theta^* \in \mathbb{R}^d$ is any critical point of function $f$, and $\alpha', \beta'$ are positive constants.*

*Proof of Lemma A.4.* When $\theta^* = \mathbf{0}$, the result follows trivially from Assumption 2.2. Assume $\|\theta^*\| > 0$. Note that

$$\langle \nabla f(\theta), \theta - \theta^* \rangle = \langle \nabla f(\theta), \theta \rangle - \langle \nabla f(\theta), \theta^* \rangle.$$

By Assumption 2.2, it holds that

$$\langle \nabla f(\theta), \theta \rangle \geq \alpha \|\theta\|^2 - \beta$$
$$\geq \alpha(\|\theta - \theta^*\|^2 + \|\theta^*\|^2 - 2\|\theta^*\|\|\theta - \theta^*\|) - \beta.$$

By Assumption 2.1, Cauchy-Schwarz inequality and triangular inequality, it holds that

$$\langle \nabla f(\theta), \theta^* \rangle \leq \|\nabla f(\theta)\|\|\theta^*\| \leq L\|\theta^*\|(1 + \|\theta - \theta^*\| + \|\theta^*\|).$$

Combing the previous two displays yields

$$\langle \nabla f(\theta), \theta - \theta^* \rangle$$
$$\geq \alpha(\|\theta - \theta^*\|^2 + \|\theta^*\|^2 - 2\|\theta^*\|\|\theta - \theta^*\|) - \beta - L\|\theta^*\|(1 + \|\theta - \theta^*\| + \|\theta^*\|)$$
$$\geq \frac{\alpha}{2}\|\theta - \theta^*\|^2 - \beta - L\|\theta^*\|^2 - L\|\theta^*\|.$$

The desired result follows by setting $\alpha' := \frac{\alpha}{2}$ and $\beta' := \beta + \left(\sqrt{\alpha} + \frac{2L}{\sqrt{\alpha}}\right)^2 \|\theta^*\|^2 + L\|\theta^*\|$. $\qquad \square$

**Lemma A.5.** *Under Assumptions 2.2 and 3.1, it holds for any $k \geq 1$ and $\theta \in \mathbb{R}^d$ that*

$$\mathbb{E}[\|\xi_{k+1}(\theta)\|^r] \leq L_\xi'^{r/4}(1 + \|\theta - \theta^*\|^r), \quad for \quad r \in \{2, 3, 4\},$$

*where $\theta^* \in \mathbb{R}^d$ is any critical point of function $f$, and $L_\xi' := 8L_\xi(1 + (\beta/\alpha)^4)$.*

*Proof of Lemma A.5.* By Assumptions 2.2 and 3.1, it holds that

$$\mathbb{E}[\|\xi_{k+1}(\theta)\|^4] \leq L_\xi(1 + \|\theta\|^4)$$
$$\leq L_\xi(1 + 8\|\theta - \theta^*\|^4 + 8\|\theta^*\|^4)$$
$$\leq L_\xi(1 + 8\|\theta - \theta^*\|^4 + 8(\beta/\alpha)^4)$$
$$\leq L_\xi'(1 + \|\theta - \theta^*\|^4),$$

where $L'_\xi := 8L_\xi(1 + (\beta/\alpha)^4)$. Similarly, for $r \in \{2, 3\}$ we have

$$
\begin{aligned}
\mathbb{E}[\|\xi_{k+1}(\theta)\|^r] &\leq \mathbb{E}[\|\xi_{k+1}(\theta)\|^4]^{r/4} \\
&\leq L_\xi^{r/4}(1 + \|\theta\|^4)^{r/4} \\
&\leq L_\xi^{r/4}(1 + \|\theta\|^r) \\
&\leq L_\xi^{r/4}(1 + 2^{r-1}\|\theta - \theta^*\|^r + 2^{r-1}\|\theta^*\|^r) \\
&\leq L_\xi^{r/4}(1 + 2^{r-1}\|\theta - \theta^*\|^r + 2^{r-1}(\beta/\alpha)^r) \\
&\leq {L'_\xi}^{r/4}(1 + \|\theta - \theta^*\|^r)\,,
\end{aligned}
$$

where $L'_\xi$ is defined above. $\qquad\square$

**Lemma A.6.** *Under Assumptions 2.1, 2.2, and 3.1, with step size $\eta < 1 \wedge \frac{1}{10\bar{L}}$, it holds for any $k \geq 0$ that*

$$
\begin{aligned}
&\mathbb{E}[\|\theta_{k+1}^{(\eta)} - \theta^*\|^4|\mathcal{F}_k] \\
&\leq (1 - 4\eta\alpha' + 32L_\dagger\eta^2)\|\theta_k^{(\eta)} - \theta^*\|^4 + \eta(4\beta' + 24\bar{L}^2 + 12{L'_\xi}^{1/2} + 64)\|\theta_k^{(\eta)} - \theta^*\|^2 \\
&\quad + \eta^2(64\bar{L}^4 + 8L'_\xi + 32(4\bar{L}^3)^2 + 32(L'_\xi)^{3/2})\,.
\end{aligned}
\tag{14}
$$

*where $L_\dagger := \bar{L}^2 + 16\left(L_\xi^{3/4}(1 + (\beta/\alpha)^3) \vee L_\xi^{1/2}(1 + (\beta/\alpha)^2) \vee L_\xi(1 + (\beta/\alpha)^4)\right)$ with $\bar{L} := L(1 + \|\theta^*\|)$, $L'_\xi$ is from Lemma A.5, and $\theta^*$ is any critical points of fuction $f$.*

*Proof of Lemma A.6.* Define $\Delta_k := \|\theta_k^{(\eta)} - \theta^*\|$. It holds by Assumption 2.1 that

$$
\|\nabla f(\theta_k^{(\eta)})\| \leq \bar{L}\Delta_k + \bar{L}\,,
$$

where $\bar{L} = L(\|\theta^*\| + 1)$. Note that

$$
\begin{aligned}
\Delta_{k+1}^4 =& (\Delta_k^2 + \eta^2\|\nabla f(\theta_k^{(\eta)}) + \xi_{k+1}(\theta_k^{(\eta)})\|^2 - 2\eta\langle\nabla f(\theta_k^{(\eta)}) + \xi_{k+1}(\theta_k^{(\eta)}), \theta_k^{(\eta)} - \theta^*\rangle)^2 \\
=& \Delta_k^4 + \eta^4\|\nabla f(\theta_k^{(\eta)}) + \xi_{k+1}(\theta_k^{(\eta)})\|^4 + 4\eta^2\langle\nabla f(\theta_k^{(\eta)}) + \xi_{k+1}(\theta_k^{(\eta)}), \theta_k^{(\eta)} - \theta^*\rangle^2 \\
& + 2\eta^2\Delta_k^2\|\nabla f(\theta_k^{(\eta)}) + \xi_{k+1}(\theta_k^{(\eta)})\|^2 - 4\eta\Delta_k^2\langle\nabla f(\theta_k^{(\eta)}) + \xi_{k+1}(\theta_k^{(\eta)}), \theta_k^{(\eta)} - \theta^*\rangle \\
& - 4\eta^3\|\nabla f(\theta_k^{(\eta)}) + \xi_{k+1}(\theta_k^{(\eta)})\|^2\langle\nabla f(\theta_k^{(\eta)}) + \xi_{k+1}(\theta_k^{(\eta)}), \theta_k^{(\eta)} - \theta^*\rangle \\
=& \Delta_k^4 + \mathrm{I} + \mathrm{II} + \mathrm{III} + \mathrm{IV} + \mathrm{V}\,,
\end{aligned}
$$

where

$$
\begin{aligned}
\mathrm{I} &:= \eta^4\|\nabla f(\theta_k^{(\eta)}) + \xi_{k+1}(\theta_k^{(\eta)})\|^4 \\
\mathrm{II} &:= 4\eta^2\langle\nabla f(\theta_k^{(\eta)}) + \xi_{k+1}(\theta_k^{(\eta)}), \theta_k^{(\eta)} - \theta^*\rangle^2 \\
\mathrm{III} &:= 2\eta^2\Delta_k^2\|\nabla f(\theta_k^{(\eta)}) + \xi_{k+1}(\theta_k^{(\eta)})\|^2 \\
\mathrm{IV} &:= -4\eta\Delta_k^2\langle\nabla f(\theta_k^{(\eta)}) + \xi_{k+1}(\theta_k^{(\eta)}), \theta_k^{(\eta)} - \theta^*\rangle \\
\mathrm{V} &:= -4\eta^3\|\nabla f(\theta_k^{(\eta)}) + \xi_{k+1}(\theta_k^{(\eta)})\|^2\langle\nabla f(\theta_k^{(\eta)}) + \xi_{k+1}(\theta_k^{(\eta)}), \theta_k^{(\eta)} - \theta^*\rangle\,.
\end{aligned}
$$

To obtain the expectation $\mathbb{E}[\Delta_{k+1}^4]$, we first calculate the conditional expectation $\mathbb{E}[\Delta_{k+1}^4|\mathcal{F}_k]$. For this, we proceed the conditional expectation of the above five terms separately. Note that

$$
\begin{aligned}
\mathbb{E}[\mathrm{I}|\mathcal{F}_k] =& \eta^4\mathbb{E}[\|\nabla f(\theta_k^{(\eta)}) + \xi_{k+1}(\theta_k^{(\eta)})\|^4|\mathcal{F}_k] \\
\leq& \eta^4\mathbb{E}[8\|\nabla f(\theta_k^{(\eta)})\|^4 + 8\|\xi_{k+1}(\theta_k^{(\eta)})\|^4|\mathcal{F}_k] \\
\leq& 8\eta^4(8\bar{L}^4\Delta_k^4 + 8\bar{L}^4 + L'_\xi\Delta_k^4 + L'_\xi)\,.
\end{aligned}
$$

The first inequality follows from the fact that $(x + y)^4 \leq 8(x^4 + y^4), \forall x, y > 0$. The last inequality follows from Assumptions 2.1 and Lemma A.5. Using the same trick and invoking Cauchy-Schwarz

inequality gives

$$\mathbb{E}[\mathrm{II}|\mathcal{F}_k] = 4\eta^2 \mathbb{E}[\langle \nabla f(\theta_k^{(\eta)}) + \xi_{k+1}(\theta_k^{(\eta)}), \, \theta_k^{(\eta)} - \theta^* \rangle^2 | \mathcal{F}_k]$$
$$\leq 4\eta^2 \Delta_k^2 \mathbb{E}[\,\|\nabla f(\theta_k^{(\eta)}) + \xi_{k+1}(\theta_k^{(\eta)})\|^2 | \mathcal{F}_k]$$
$$\leq 8\eta^2 \Delta_k^2 (2\bar{L}^2 \Delta_k^2 + 2\bar{L}^2 + L_\xi'^{\,1/2}\Delta_k^2 + L_\xi'^{\,1/2}) \,.$$

Similarly, we have

$$\mathbb{E}[\mathrm{III}|\mathcal{F}_k] = 2\eta^2 \Delta_k^2 \mathbb{E}[\,\|\nabla f(\theta_k^{(\eta)}) + \xi_{k+1}(\theta_k^{(\eta)})\|^2 | \mathcal{F}_k]$$
$$\leq 4\eta^2 \Delta_k^2 (2\bar{L}^2 \Delta_k^2 + 2\bar{L}^2 + L_\xi'^{\,1/2}\Delta_k^2 + L_\xi'^{\,1/2}) \,.$$

Using Cauchy-Schwarz inequality again, we obtain

$$\mathbb{E}[\mathrm{V}|\mathcal{F}_k] = \mathbb{E}[-4\eta^3 \,\|\nabla f(\theta_k^{(\eta)}) + \xi_{k+1}(\theta_k^{(\eta)})\|^2 \langle \nabla f(\theta_k^{(\eta)}) + \xi_{k+1}(\theta_k^{(\eta)}), \, \theta_k^{(\eta)} - \theta^* \rangle | \mathcal{F}_k]$$
$$\leq 4\eta^3 \mathbb{E}[\|\nabla f(\theta_k^{(\eta)}) + \xi_{k+1}(\theta_k^{(\eta)})\|^3 \|\theta_k^{(\eta)} - \theta^*\| | \mathcal{F}_k]$$
$$= 4\eta^3 \Delta_k \mathbb{E}\big[\|\nabla f(\theta_k^{(\eta)}) + \xi_{k+1}(\theta_k^{(\eta)})\|^3 | \mathcal{F}_k\big]$$
$$\leq 4\eta^3 \Delta_k \mathbb{E}\big[4\|\nabla f(\theta_k^{(\eta)})\|^3 + 4\|\xi_{k+1}(\theta_k^{(\eta)})\|^3 | \mathcal{F}_k\big] \,.$$

Note that by Lemma A.5, it holds for any $k \geq 1$ and $\theta \in \mathbb{R}^d$ that

$$\mathbb{E}[\|\xi_k(\theta)\|^3] \leq L_\xi'^{\,3/4}(1 + \|\theta - \theta^*\|^3) \,.$$

Combining this with the previous display yields

$$\mathbb{E}[\mathrm{V}|\mathcal{F}_k] \leq 16\eta^3 \Delta_k (4\bar{L}^3 \Delta_k^3 + 4\bar{L}^3 + L_\xi'^{\,3/4} + L_\xi'^{\,3/4}\Delta_k^3)$$
$$= 64\bar{L}^3 \eta^3 \Delta_k^4 + 16\eta^3 L_\xi'^{\,3/4}\Delta_k^4 + 16\eta^2 (\Delta_k \eta 4\bar{L}^3 + \Delta_k \eta L_\xi'^{\,3/4}) \,.$$

Collecting pieces gives

$$\mathbb{E}[\Delta_{k+1}^4 | \mathcal{F}_k] \leq \Delta_k^4 (1 + 64\eta^4 \bar{L}^4 + 64\eta^3 \bar{L}^3 + 24\eta^2 \bar{L}^2 + 8\eta^2 L_\xi' + 12\eta^2 L_\xi'^{\,1/2} + 16\eta^2 L_\xi'^{\,3/4})$$
$$- 4\eta \Delta_k^2 \langle \nabla f(\theta_k^{(\eta)}), \, \theta_k^{(\eta)} - \theta^* \rangle$$
$$+ \eta^2 \big(64\eta^2 \bar{L}^4 + 8\eta^2 L_\xi' + 24\bar{L}^2 \Delta_k^2 + 12 L_\xi' \Delta_k^2 + 64\Delta_k^2 + 32(\eta 4\bar{L}^3)^2 + 32(\eta L_\xi'^{\,3/4})^2\big)$$
$$\leq \Delta_k^4 [1 + 32\eta^2 (\bar{L}^2 + L_\xi' + L_\xi'^{\,1/2} + L_\xi'^{\,3/4})] - 4\eta \Delta_k^2 \langle \nabla f(\theta_k^{(\eta)}), \, \theta_k^{(\eta)} - \theta^* \rangle$$
$$+ \eta \big(64\bar{L}^4 \eta + 8 L_\xi' \eta + 24\bar{L}^2 \Delta_k^2 + 12 L_\xi'^{\,1/2}\Delta_k^2 + 64\Delta_k^2 + 32(4\bar{L}^3)^2 \eta + 32(L_\xi')^{3/2}\eta\big) \,.$$

The above inequalities are based on the fact that $\eta < \frac{1}{10\bar{L}} \wedge 1$ and $xy \leq 2x^2 + 2y^2, \forall x, y > 0$. By Lemma A.4, we handle the term IV as following

$$\mathbb{E}[\Delta_{k+1}^4 | \mathcal{F}_k] \leq \Delta_k^4 \big(1 - 4\eta \alpha' + 32\eta^2 (\bar{L}^2 + L_\xi' + L_\xi'^{\,1/2} + L_\xi'^{\,3/4})\big)$$
$$+ \eta \big(4\beta' \Delta_k^2 + 64\bar{L}^4 \eta + 8 L_\xi' \eta + 24\bar{L}^2 \Delta_k^2 + 12 L_\xi'^{\,1/2}\Delta_k^2 + 64\Delta_k^2 + 32(4\bar{L}^3)^2 \eta + 32 L_\xi'^{\,3/2}\eta\big) \,.$$

Define $L_\dagger := \bar{L}^2 + 16\Big(L_\xi'^{\,3/4}(1 + (\beta/\alpha)^3) \vee L_\xi'^{\,1/2}(1 + (\beta/\alpha)^2) \vee L_\xi(1 + (\beta/\alpha)^4)\Big)$. Note that $L_\dagger > \bar{L}^2 + L_\xi' + L_\xi'^{\,1/2} + L_\xi'^{\,3/4}$. Combing this with the previous display gives

$$\mathbb{E}[\Delta_{k+1}^4 | \mathcal{F}_k]$$
$$\leq \Delta_k^4 (1 - 4\eta \alpha' + 32\eta^2 L_\dagger)$$
$$+ \eta \big(4\beta' \Delta_k^2 + 64\bar{L}^4 \eta + 8 L_\xi' \eta + 24\bar{L}^2 \Delta_k^2 + 12 L_\xi'^{\,1/2}\Delta_k^2 + 64\Delta_k^2 + 32(4\bar{L}^3)^2 \eta + 32 L_\xi'^{\,3/2}\eta\big)$$
$$\leq (1 - 4\eta \alpha' + 32 L_\dagger \eta^2)\Delta_k^4 + \eta(4\beta' + 24\bar{L}^2 + 12 L_\xi'^{\,1/2} + 64)\Delta_k^2 + \eta^2(64\bar{L}^4 + 8 L_\xi' + 32(4\bar{L}^3)^2 + 32 L_\xi'^{\,3/2}) \,.$$

$\square$

**Lemma A.7.** *Assume Assumptions 2.1-2.3 holds. With the step size*

$$\eta \leq \frac{\alpha - \sqrt{(\alpha^2 - (3L^2 + L_\xi)) \vee 0}}{3L^2 + L_\xi} \wedge \frac{\alpha}{64L_\dagger} \wedge 1 \,,$$

*the chain (2) has the stationary distribution $\pi_\eta$, and the chain has finite 4-th moment:*

$$\mathbb{E}[\|\theta_{k+1}^{(\eta)}\|^4] \leq \mu_{4,\eta} \,,$$

*where*

$$\mu_{4,\eta} := \frac{8}{7\alpha}\Big((\beta + 6L^2 + 3L_\xi^{1/2} + 16)\mu_{2,\eta} + 16L^4 + 2L_\xi + 128L^6 + 8L_\xi^{3/2}\Big)$$

*with $\mu_{2,\eta}$ defined in Corollary A.1, and $L_\dagger$ defined in Lemma A.6.*

*Proof of Lemma A.7.* Similar to display (14), we can derive

$$\mathbb{E}[\|\theta_{k+1}^{(\eta)}\|^4|\mathcal{F}_k] \leq (1 - 4\eta\alpha + 32L_0^\dagger\eta^2)\|\theta_k^{(\eta)}\|^4$$
$$+ \eta\big[(4\beta + 24L^2 + 12L_\xi^{1/2} + 64)\|\theta_k^{(\eta)}\|^2 + \eta(64L^4 + 8L_\xi + 32(4L^3)^2 + 32L_\xi^{3/2})\big]\,,$$

where $L_0^\dagger := L^2 + L_\xi + L_\xi^{1/2} + L_\xi^{3/4}$. Recall the definition of $L_\dagger$ in Lemma A.6, it holds that $L_\dagger \geq L_0^\dagger$, which implies

$$\mathbb{E}[\|\theta_{k+1}^{(\eta)}\|^4|\mathcal{F}_k] \leq (1 - 4\eta\alpha + 32L_\dagger\eta^2)\|\theta_k^{(\eta)}\|^4$$
$$+ \eta\big[(4\beta + 24L^2 + 12L_\xi^{1/2} + 64)\|\theta_k^{(\eta)}\|^2 + \eta(64L^4 + 8L_\xi + 32(4L^3)^2 + 32L_\xi^{3/2})\big]\,,$$

Note that the chain starts from the stationary distribution $\pi_\eta$, taking the expectation on both sides gives

$$(4\eta\alpha - 32L_\dagger\eta^2)\mathbb{E}[\|\theta_k^{(\eta)}\|^4]$$
$$\leq \eta(4\beta + 24L^2 + 12L_\xi^{1/2} + 64)\mathbb{E}[\|\theta_k^{(\eta)}\|^2] + \eta^2(64L^4 + 8L_\xi + 32(4L^3)^2 + 32L_\xi^{3/2})\,.$$

We also note that $\mathbb{E}[\|\theta_k^{(\eta)}\|^2] = \mu_{2,\eta}$ for $\mu_{2,\eta}$ from Corollary A.1. Plugging this into the previous display and rearranging the inequality yields

$$\mathbb{E}[\|\theta_{k+1}^{(\eta)}\|^4]$$
$$\leq \frac{\eta}{4\eta\alpha - 32L_\dagger\eta^2}(4\beta + 24L^2 + 12L_\xi^{1/2} + 64)\mu_{2,\eta} + \frac{\eta^2}{4\eta\alpha - 32L_\dagger\eta^2}(64L^4 + 8L_\xi + 32(4L^3)^2 + 32L_\xi^{3/2})$$
$$\leq \frac{\eta}{4\eta\alpha - 32L_\dagger\eta^2}(4\beta + 24L^2 + 12L_\xi^{1/2} + 64)\mu_{2,\eta} + \frac{\eta}{4\eta\alpha - 32L_\dagger\eta^2}(64L^4 + 8L_\xi + 32(4L^3)^2 + 32L_\xi^{3/2})$$
$$\leq \frac{2}{7\alpha}\Big[(4\beta + 24L^2 + 12L_\xi^{1/2} + 64)\mu_{2,\eta} + (64L^4 + 8L_\xi + 32(4L^3)^2 + 32L_\xi^{3/2})\Big]$$

as desired. $\square$

We are now ready to prove Proposition B.1.

*Proof of Proposition B.1.* Define $\Delta_k := \|\theta_k^{(\eta)} - \theta^*\|$. By Lemma A.6, we have

$$\mathbb{E}[\Delta_{k+1}^4|\mathcal{F}_k]$$
$$\leq (1 - 4\eta\alpha' + 32L_\dagger\eta^2)\Delta_k^4 + \eta(4\beta' + 24\bar{L}^2 + 12L_\xi'^{1/2} + 64)\Delta_k^2$$
$$+ \eta^2(64\bar{L}^4 + 8L_\xi' + 32(4\bar{L}^3)^2 + 32L_\xi'^{3/2})\,.$$

Taking expectation on both sides then gives

$$\mathbb{E}[\Delta_{k+1}^4]$$
$$\leq (1 - 4\eta\alpha' + 32L_\dagger\eta^2)\mathbb{E}[\Delta_k^4] + \eta(4\beta' + 24\bar{L}^2 + 12L_\xi'^{1/2} + 64)\mathbb{E}[\Delta_k^2]$$
$$+ \eta^2(64\bar{L}^4 + 8L_\xi' + 32(4\bar{L}^3)^2 + 32L_\xi'^{3/2})\,.$$

Set

$$\varrho := 1 - 4\eta\alpha' + 32L_\dagger\eta^2$$

$$A_1 := 64\bar{L}^4 + 8L'_\xi + 32(4\bar{L}^3)^2 + 32L'^{3/2}_\xi$$

$$A_2 := 4\beta' + 24\bar{L}^2 + 12L'^{1/2}_\xi + 64.$$

By Cauchy-Schwatz inequality, we then have

$$\mathbb{E}[\Delta^4_{k+1}] \leq \varrho\mathbb{E}[\Delta^4_k] + A_1\eta^2 + A_2\mathbb{E}^{1/2}[\Delta^4_k]\eta.$$

Note that when $0 < \eta < \frac{\alpha' - \sqrt{(\alpha'^2 - 4L_\dagger)}}{16L_\dagger}\mathbb{1}(\alpha'^2 > 8L_\dagger) + \frac{\alpha'}{32L_\dagger}\mathbb{1}(\alpha'^2 \leq 8L_\dagger)$, it follows that

$$\varrho > \frac{1}{2}\mathbb{1}(\alpha'^2 \geq 8L_\dagger) + (1 - \frac{3\alpha'^2}{32L^2_\dagger})\mathbb{1}(\alpha'^2 < 8L_\dagger) \geq \frac{1}{4}.$$

Set $D := \sqrt{A_1} \vee A_2$. We then find

$$\mathbb{E}^{1/2}[\Delta^4_{k+1}] \leq \sqrt{\varrho}\,\mathbb{E}^{1/2}[\Delta^4_k] + D\eta.$$

By a straightforward induction, we have

$$\mathbb{E}^{1/2}[\Delta^4_k] \leq \varrho^{k/2}\mathbb{E}^{1/2}[\Delta^4_0] + \frac{D\eta}{1 - \sqrt{\varrho}}.$$

Notice that $\eta \leq \frac{\alpha'}{16L_\dagger}$, it then follows that

$$\varrho = 1 - 4\eta\alpha' + 32L_\dagger\eta^2 \leq 1 - 2\eta\alpha',$$

which implies

$$\frac{1}{1 - \sqrt{\varrho}} \leq \frac{1}{1 - \sqrt{1 - 2\eta\alpha'}} \leq \frac{1}{\eta\alpha'}.$$

Combining this with previous display gives

$$\mathbb{E}^{1/2}[\Delta^4_k] \leq \varrho^{k/2}\mathbb{E}^{1/2}[\Delta^4_0] + \frac{D}{\alpha'}.$$

By Proposition 2.1, there exists a unique stationary distribution $\pi_\eta$.

Consider the chain starting from the stationary distribution $\pi_\eta$. Note that $\mathbb{E}[\Delta^4_0] \leq 8(\mathbb{E}[\|\theta^{(\eta)}_0\|^4] + \|\theta^*\|^4)$. By Lemma A.7, it follows that

$$\mathbb{E}[\Delta^4_0] \leq 8\mu_{4,\eta} + 8\|\theta^*\|^4,$$

where the constant $\mu_{4,\eta}$ is defined in Lemma A.7. Plugging this into previous display provides us with

$$\left(\int \|\theta - \theta^*\|^4\pi_\eta(d\theta)\right)^{1/4} = \mathcal{O}(1).$$

Note that it holds for the $L_\phi$-Lipschitz continuous test function $\phi$ that

$$|\pi_\eta(\phi) - \phi(\theta^*)| \leq L_\phi \int \|\theta - \theta^*\|\pi_\eta(d\theta)$$

$$\leq L_\phi\left[\int \|\theta - \theta^*\|^4\pi_\eta(d\theta)\right]^{1/4},$$

Thus, we obtain

$$|\pi_\eta(\phi) - \phi(\theta^*)| = \mathcal{O}(1)$$

as desired.

□

To prove Theorem 3.1, we need the following lemma.

**Lemma A.8.** *For any $a, b, \delta > 0$, it holds for any $x \geq \frac{\delta}{a} + \sqrt{\frac{b}{a}}$ that*

$$ax^2 - b \geq \delta x \,.$$

*Proof of Lemma A.8.* Define the function $h(x) := ax^2 - b - \delta x$. When $x \geq \frac{\delta + \sqrt{\delta^2 + 4ab}}{2a}$, it holds that $h(x) \geq 0$. Note that $\sqrt{\delta^2 + 4ab} \leq \delta + \sqrt{4ab}$, it follows that when

$$x \geq \frac{\delta + \delta + \sqrt{4ab}}{2a} \,,$$

it holds that $h(x) \geq 0$. The desired result then follows readily. $\qquad\square$

Now, we are ready to prove Theorem 3.1.

*Proof of Theorem 3.1.* Consider the chain $\{\theta_k^{(\eta)}\}_{k \geq 0}$ starting from the stationary distribution $\pi_\eta$. Define $\Delta_k := \|\theta_k^{(\eta)} - \theta^*\|$. Note that under Assumptions 3.2 and 3.1, Lemma A.5 still holds. By Assumptions 2.1, 3.1, and Lemma A.5 , we have

$$\mathbb{E}\big[\Delta_{k+1}^2 | \mathcal{F}_k\big]$$
$$=\mathbb{E}\big[\Delta_k^2 + \eta^2 \|\nabla f(\theta_k^{(\eta)})\|^2 + \eta^2 \|\xi_{k+1}(\theta_k^{(\eta)})\|^2 - 2\eta\langle \nabla f(\theta_k^{(\eta)}), \theta_k^{(\eta)} - \theta^*\rangle | \mathcal{F}_k\big]$$
$$\leq\Delta_k^2 + \eta^2\big(3L^2(2\Delta_k^2 + 2\|\theta^*\|^2 + 3) + {L'_\xi}^{1/2}(1 + \Delta_k^2)\big) - 2\eta\langle \nabla f(\theta_k^{(\eta)}), \theta_k^{(\eta)} - \theta^*\rangle$$
$$=\Delta_k^2 + 6L^2\eta^2\Delta_k^2 + {L'_\xi}^{1/2}\eta^2\Delta_k^2 + \eta^2 C_1 - 2\eta\langle \nabla f(\theta_k^{(\eta)}), \theta_k^{(\eta)} - \theta^*\rangle$$

where $C_1 := 6\|\theta^*\|^2 L^2 + 9L^2 + {L'_\xi}^{1/2}$. Note that the chain starts from the stationary distribution $\pi_\eta$, which implies $\mathbb{E}[\Delta_{k+1}^2] = \mathbb{E}[\Delta_k^2]$ for all $k \geq 0$. Taking the expectation on both sides and rearranging the inequality yields

$$\mathbb{E}[\langle \nabla f(\theta_k^{(\eta)}), \theta_k^{(\eta)} - \theta^*\rangle] \leq \eta(3L^2 + {L'_\xi}^{1/2})\mathbb{E}[\Delta_k^2] + \frac{\eta}{2}C_1 \,.$$

By Corollary A.1, it follows that

$$\mathbb{E}[\langle \nabla f(\theta_k^{(\eta)}), \theta_k^{(\eta)} - \theta^*\rangle] \leq C_2\eta \,, \tag{15}$$

where $C_2 := 2(3L^2 + {L'_\xi}^{1/2})(\mu_{2,\eta} + \|\theta^*\|^2) + C_1/2$ and $\mu_{2,\eta}$ is defined in Corollary A.1. Moreover, by Assumption 3.2, Lemma A.8, and Jensen's inequality, we have

$$\mathbb{E}[\langle \nabla f(\theta_k^{(\eta)}), \theta_k^{(\eta)} - \theta^*\rangle] \geq \delta\mathbb{E}[\Delta_k \mathbb{1}(\Delta_k \geq R)] + g(\mathbb{E}[\Delta_k \mathbb{1}(\Delta_k < R)]) \,.$$

Combining this with previous display provides us with

$$\mathbb{E}[\Delta_k \mathbb{1}(\Delta_k \geq R)] \leq \frac{C_2}{\delta}\eta \,,$$

and

$$\mathbb{E}[\Delta_k \mathbb{1}(\Delta_k < R)] \leq g^{-1}(C_2\eta) \,.$$

Collecting pieces then gives

$$\mathbb{E}\big[\Delta_k\big] =\mathbb{E}\big[\Delta_k \mathbb{1}(\|\theta_k^{(\eta)} - \theta^*\| < R)\big] + \mathbb{E}\big[\Delta_k \mathbb{1}(\|\theta_k^{(\eta)} - \theta^*\| \geq R)\big]$$
$$\leq\frac{C_2}{\delta}\eta + g^{-1}(C_2\eta) \,.$$

Thus, it holds for the $L_\phi$-Lipschitz continuous test function $\phi$ that

$$|\pi_\eta(\phi) - \phi(\theta^*)| \leq L_\phi \int \|\theta - \theta^*\| \pi_\eta(d\theta) \leq L_\phi\Big(\frac{C_2}{\delta}\eta + g^{-1}(C_2\eta)\Big) \,.$$

$$\square$$

Now, we provide the proof of Theorem 3.2.

*Proof of Theorem 3.2.* Consider the chain $\{\theta_k^{(\eta)}\}_{k\geq 0}$ starting from the stationary distribution $\pi_\eta$. Note that by the assumption that $\|\nabla^2 f(\theta)\| \leq \tilde{L}(1 + \|\theta\|)$ and Taylor expansion, we have

$$
\begin{aligned}
f(\theta_{k+1}^{(\eta)}) =& f(\theta_k^{(\eta)}) + \langle \nabla f(\theta_k^{(\eta)}), \theta_{k+1}^{(\eta)} - \theta_k^{(\eta)} \rangle + \frac{1}{2}(\theta_{k+1}^{(\eta)} - \theta_k^{(\eta)})^\top \nabla^2 f(\tilde{\theta})(\theta_{k+1}^{(\eta)} - \theta_k^{(\eta)}) \\
\leq& f(\theta_k^{(\eta)}) + \langle \nabla f(\theta_k^{(\eta)}), \theta_{k+1}^{(\eta)} - \theta_k^{(\eta)} \rangle + \frac{1}{2}\tilde{L}\|\theta_{k+1}^{(\eta)} - \theta_k^{(\eta)}\|^2(1 + \|\tilde{\theta}\|),
\end{aligned}
$$

where $\tilde{\theta} \in \mathbb{R}^d$ is a convex combination between $\theta_{k+1}^{(\eta)}$ and $\theta_k^{(\eta)}$. By definition of SGD iterates in (2), it follows that

$$
\begin{aligned}
f(\theta_{k+1}^{(\eta)}) \leq& f(\theta_k^{(\eta)}) - \eta\langle \nabla f(\theta_k^{(\eta)}), \nabla f(\theta_k^{(\eta)}) + \xi_{k+1}(\theta_k^{(\eta)}) \rangle + \frac{\tilde{L}}{2}\eta^2\|\nabla f(\theta_k^{(\eta)}) + \xi_{k+1}(\theta_k^{(\eta)})\|^2(1 + \|\tilde{\theta}\|) \\
=& f(\theta_k^{(\eta)}) - \eta\langle \nabla f(\theta_k^{(\eta)}), \nabla f(\theta_k^{(\eta)}) + \xi_{k+1}(\theta_k^{(\eta)}) \rangle \\
&+ \frac{\tilde{L}}{2}\eta^2\big(\|\nabla f(\theta_k^{(\eta)})\|^2 + \|\xi_{k+1}(\theta_k^{(\eta)})\|^2 + 2\langle \nabla f(\theta_k^{(\eta)}), \xi_{k+1}(\theta_k^{(\eta)}) \rangle\big)(1 + \|\tilde{\theta}\|) \\
\leq& f(\theta_k^{(\eta)}) - \eta\langle \nabla f(\theta_k^{(\eta)}), \nabla f(\theta_k^{(\eta)}) + \xi_{k+1}(\theta_k^{(\eta)}) \rangle \\
&+ \frac{\tilde{L}}{2}\eta^2\big(\|\nabla f(\theta_k^{(\eta)})\|^2 + \|\xi_{k+1}(\theta_k^{(\eta)})\|^2 + 2\langle \nabla f(\theta_k^{(\eta)}), \xi_{k+1}(\theta_k^{(\eta)}) \rangle\big)(1 + \max\{\|\theta_k^{(\eta)}\|, \|\theta_{k+1}^{(\eta)}\|\}) \\
\leq& f(\theta_k^{(\eta)}) - \eta\|\nabla f(\theta_k^{(\eta)})\|^2 - \eta\langle \nabla f(\theta_k^{(\eta)}), \xi_{k+1}(\theta_k^{(\eta)}) \rangle \\
&+ \frac{\tilde{L}}{2}\eta^2\big(\|\nabla f(\theta_k^{(\eta)})\|^2 + \|\xi_{k+1}(\theta_k^{(\eta)})\|^2 + 2\langle \nabla f(\theta_k^{(\eta)}), \xi_{k+1}(\theta_k^{(\eta)}) \rangle\big) \\
&+ \frac{\tilde{L}}{2}\eta^2\big(\|\nabla f(\theta_k^{(\eta)})\|^2 + \|\xi_{k+1}(\theta_k^{(\eta)})\|^2 + 2\langle \nabla f(\theta_k^{(\eta)}), \xi_{k+1}(\theta_k^{(\eta)}) \rangle\big)(\|\theta_k^{(\eta)}\| + \|\theta_{k+1}^{(\eta)}\|).
\end{aligned}
$$

Taking the conditional expectation on both sides, using Cauchy-Schwarz inequality, Assumption 3.1 and the fact that $(1 + x^4)^{1/2} \leq 1 + x^2, \forall x > 0$ gives

$$
\begin{aligned}
&\mathbb{E}[f(\theta_{k+1}^{(\eta)})|\mathcal{F}_k] \\
\leq& f(\theta_k^{(\eta)}) + (\frac{\tilde{L}}{2}\eta^2 - \eta)\|\nabla f(\theta_k^{(\eta)})\|^2 + \frac{\tilde{L}}{2}L_\xi\eta^2(1 + \|\theta_k^{(\eta)}\|^2) + 0 \\
&+ \frac{\tilde{L}}{2}\eta^2\mathbb{E}\Big[\|\nabla f(\theta_k^{(\eta)})\|^2(\|\theta_k^{(\eta)}\| + \|\theta_{k+1}^{(\eta)}\|)|\mathcal{F}_k\Big] \\
&+ \frac{\tilde{L}}{2}\eta^2\mathbb{E}^{1/2}\Big[\|\xi_{k+1}(\theta_k^{(\eta)})\|^4|\mathcal{F}_k\Big]\mathbb{E}^{1/2}\Big[(\|\theta_k^{(\eta)}\| + \|\theta_{k+1}^{(\eta)}\|)^2|\mathcal{F}_k\Big] \\
&+ 0 + \tilde{L}\eta^2\mathbb{E}[\|\nabla f(\theta_k^{(\eta)})\|\|\xi_{k+1}(\theta_k^{(\eta)})\|\|\theta_{k+1}^{(\eta)}\||\mathcal{F}_k] \\
\leq& f(\theta_k^{(\eta)}) + (\frac{\tilde{L}}{2}\eta^2 - \eta)\|\nabla f(\theta_k^{(\eta)})\|^2 + \frac{\tilde{L}}{2}L_\xi\eta^2(1 + \|\theta_k^{(\eta)}\|^2) \\
&+ \frac{\tilde{L}}{2}\eta^2\mathbb{E}\Big[\|\nabla f(\theta_k^{(\eta)})\|^2(\|\theta_k^{(\eta)}\| + \|\theta_{k+1}^{(\eta)}\|)|\mathcal{F}_k\Big] \\
&+ \frac{\tilde{L}}{2}\eta^2 L_\xi^{1/2}(1 + \|\theta_k^{(\eta)}\|^2)\mathbb{E}^{1/2}\Big[(\|\theta_k^{(\eta)}\| + \|\theta_{k+1}^{(\eta)}\|)^2|\mathcal{F}_k\Big] \\
&+ \tilde{L}\eta^2\mathbb{E}[\|\nabla f(\theta_k^{(\eta)})\|\|\xi_{k+1}(\theta_k^{(\eta)})\|\|\theta_{k+1}^{(\eta)}\||\mathcal{F}_k].
\end{aligned}
$$

We then take expectation on both sides. For this, we bound the last three terms separately. Note that the chain starts from the initial distribution $\pi_\eta$. By Hölder's inequality, we obtain

$$
\begin{aligned}
&\mathbb{E}[\|\nabla f(\theta_k^{(\eta)})\|^2(\|\theta_k^{(\eta)}\| + \|\theta_{k+1}^{(\eta)}\|)] \\
\leq& \mathbb{E}[\|\nabla f(\theta_k^{(\eta)})\|^2\|\theta_k^{(\eta)}\|] + \mathbb{E}[\|\nabla f(\theta_k^{(\eta)})\|^2\|\theta_{k+1}^{(\eta)}\|] \\
\leq& \mathbb{E}^{1/2}[\|\nabla f(\theta_k^{(\eta)})\|^4]\mathbb{E}^{1/2}[\|\theta_k^{(\eta)}\|^2] + \mathbb{E}^{1/2}[\|\nabla f(\theta_k^{(\eta)})\|^4]\mathbb{E}^{1/2}[\|\theta_k^{(\eta)}\|^2].
\end{aligned}
$$

By Assumption 2.1 and the fact that $(x + y)^4 \leq 9(x^4 + y^4), \forall x, y \in \mathbb{R}$, we have

$$\mathbb{E}^{1/2}[\|\nabla f(\theta_k^{(\eta)})\|^4] \leq L^2 \mathbb{E}^{1/2}[(1 + \|\theta_k^{(\eta)}\|)^4] \leq 3L^2 \sqrt{1 + \mathbb{E}[\|\theta_k^{(\eta)}\|^4]} \,.$$

By Lemma A.7, it holds that $\mathbb{E}[\|\theta_k^{(\eta)}\|^4] < \mu_{4,\eta}$, where the constant $\mu_{4,\eta}$ is defined in Lemma A.7. Moreover, by Corollary A.1, we also have $\mathbb{E}[\|\theta_k^{(\eta)}\|^2] \leq \mu_{2,\eta}$, where the constant $\mu_{2,\eta}$ is defined in Corollary A.1. Combining these with previous display gives

$$\mathbb{E}[\|\nabla f(\theta_k^{(\eta)})\|^2 (\|\theta_k^{(\eta)}\| + \|\theta_{k+1}^{(\eta)}\|)] \leq 6L^2 \sqrt{1 + \mu_{4,\eta}} \sqrt{\mu_{2,\eta}} \,.$$

Using the same trick, we obtain

$$\mathbb{E}\left[ (1 + \|\theta_k^{(\eta)}\|^2) \mathbb{E}^{1/2}[(\|\theta_k^{(\eta)}\| + \|\theta_{k+1}^{(\eta)}\|)^2 | \mathcal{F}_k] \right]$$
$$\leq \mathbb{E}^{1/2}[(1 + \|\theta_k^{(\eta)}\|^2)^2] \mathbb{E}^{1/2}\left[ \mathbb{E}[(\|\theta_k^{(\eta)}\| + \|\theta_{k+1}^{(\eta)}\|)^2 | \mathcal{F}_k] \right]$$
$$\leq \mathbb{E}^{1/2}[2 + 2\|\theta_k^{(\eta)}\|^4] \mathbb{E}^{1/2}\left[ \mathbb{E}[2\|\theta_k^{(\eta)}\|^2 + 2\|\theta_{k+1}^{(\eta)}\|^2 | \mathcal{F}_k] \right]$$
$$\leq 4 \mathbb{E}^{1/2}[1 + \|\theta_k^{(\eta)}\|^4] \mathbb{E}^{1/2}[\|\theta_k^{(\eta)}\|^2]$$
$$\leq 4 \sqrt{\mu_{2,\eta}} \sqrt{1 + \mu_{4,\eta}} \,.$$

By Assumptions 2.1 and 3.1, we have

$$\mathbb{E}\left[ \mathbb{E}[\|\nabla f(\theta_k^{(\eta)})\| \|\xi_{k+1}(\theta_k^{(\eta)})\| \|\theta_{k+1}^{(\eta)}\| | \mathcal{F}_k] \right]$$
$$\leq \mathbb{E}\left[ \|\nabla f(\theta_k^{(\eta)})\| \mathbb{E}^{1/2}[\|\xi_{k+1}(\theta_k^{(\eta)})\|^2 | \mathcal{F}_k] \mathbb{E}^{1/2}[\|\theta_{k+1}^{(\eta)}\|^2 | \mathcal{F}_k] \right]$$
$$\leq L_\xi^{1/2} L \mathbb{E}\left[ (1 + \|\theta_k^{(\eta)}\|)(1 + \|\theta_k^{(\eta)}\|^2)^{1/2} \mathbb{E}^{1/2}[\|\theta_{k+1}^{(\eta)}\|^2 | \mathcal{F}_k] \right]$$
$$\leq L L_\xi^{1/2} \mathbb{E}\left[ (1 + \|\theta_k^{(\eta)}\|)^2 \mathbb{E}^{1/2}[\|\theta_{k+1}^{(\eta)}\|^2 | \mathcal{F}_k] \right]$$
$$\leq L L_\xi^{1/2} \mathbb{E}^{1/2}[(1 + \|\theta_k^{(\eta)}\|)^4] \mathbb{E}^{1/4}[\|\theta_k^{(\eta)}\|^4]$$
$$\leq L L_\xi^{1/2} \sqrt{8 + 8\mu_{4,\eta}} (\mu_{4,\eta})^{1/4}$$
$$= 3 L L_\xi^{1/2} (\mu_{4,\eta} + \mu_{4,\eta}^{3/4}) \,.$$

Collecting pieces then gives

$$\mathbb{E}[f(\theta_{k+1}^{(\eta)})]$$
$$\leq \mathbb{E}[f(\theta_k^{(\eta)})] + (\frac{\tilde{L}}{2}\eta^2 - \eta) \mathbb{E}[\|\nabla f(\theta_k^{(\eta)})\|^2] + \tilde{L} L_\xi \eta^2 (1 + \mu_{2,\eta})$$
$$\quad + 3 \tilde{L} L^2 \eta^2 \mu_{2,\eta}^{1/2} \sqrt{1 + \mu_{4,\eta}} + 2 \tilde{L} L_\xi^{1/2} \eta^2 \mu_{2,\eta}^{1/2} \sqrt{1 + \mu_{4,\eta}} + 3 \tilde{L} L L_\xi^{1/2} \eta^2 (\mu_{4,\eta} + \mu_{4,\eta}^{3/4})$$
$$\leq \mathbb{E}[f(\theta_k^{(\eta)})] + (\frac{\tilde{L}}{2}\eta^2 - \eta) \mathbb{E}[\|\nabla f(\theta_k^{(\eta)})\|^2] + 12\eta^2 \tilde{L}(L + L_\xi^{1/2} + L_\xi^{1/4})^2 \left( 1 + \mu_{2,\eta} + \mu_{4,\eta} + \mu_{4,\eta}^{3/4} \right) \,.$$

Recall that the iterates $\{\theta_k^{(\eta)}\}_{k \geq 0}$ starts from the stationary distribution $\pi_\eta$ and $\eta < \frac{2}{\tilde{L}}$. Rearranging the above display gives

$$\mathbb{E}[\|\nabla f(\theta_k^{(\eta)})\|^2] \leq \frac{2\tilde{M}\eta}{2 - \tilde{L}\eta} \,,$$

where

$$\tilde{M} := 12\tilde{L}(L + L_\xi^{1/2} + L_\xi^{1/4})^2 \left( 1 + \mu_{2,\eta} + \mu_{4,\eta} + \mu_{4,\eta}^{3/4} \right) \,.$$

By Assumption 3.3 and Jensen's inequality, it holds that

$$\mathbb{E}[\|\nabla f(\theta_k^{(\eta)})\|^2] \geq \mathbb{E}[g(f(\theta_k^{(\eta)}) - f^*)\mathbb{1}(\|\theta_k^{(\eta)} - \theta^*\| \leq R)] + \gamma \mathbb{E}[(f(\theta_k^{(\eta)}) - f^*)\mathbb{1}(\|\theta_k^{(\eta)} - \theta^*\| > R)]$$
$$\geq g(\mathbb{E}[(f(\theta_k^{(\eta)}) - f^*)\mathbb{1}(\|\theta_k^{(\eta)} - \theta^*\| \leq R)]) + \gamma \mathbb{E}[(f(\theta_k^{(\eta)}) - f^*)\mathbb{1}(\|\theta_k^{(\eta)} - \theta^*\| > R)] \,.$$

Combing this with previous display gives

$$0 \leq \mathbb{E}[(f(\theta_k^{(\eta)})] - f^*)\mathbb{1}(\|\theta_k^{(\eta)} - \theta^*\| \leq R)] \leq g^{-1}\Big(\frac{2\tilde{M}\eta}{2 - \tilde{L}\eta}\Big)$$

$$0 \leq \mathbb{E}[(f(\theta_k^{(\eta)})] - f^*)\mathbb{1}(\|\theta_k^{(\eta)} - \theta^*\| > R)] \leq \frac{2\tilde{M}\eta}{2 - \tilde{L}\eta}\,.$$

This implies

$$
\begin{aligned}
0 \leq \pi_\eta(f) - f^* &= \mathbb{E}[(f(\theta_k^{(\eta)})] - f^* \\
&= \mathbb{E}[(f(\theta_k^{(\eta)}) - f^*)\mathbb{1}(\|\theta_k^{(\eta)} - \theta^*\| \leq R)] + \mathbb{E}[(f(\theta_k^{(\eta)}) - f^*)\mathbb{1}(\|\theta_k^{(\eta)} - \theta^*\| > R)] \\
&\leq g^{-1}\Big(\frac{2\tilde{M}\eta}{2 - \tilde{L}\eta}\Big) + \frac{2\tilde{M}\eta}{2 - \tilde{L}\eta}\,.
\end{aligned}
$$

When the test function $\phi$ satisfies $\phi = \tilde{\phi} \circ f$ with the $L_{\tilde{\phi}}$-Lipschitz function $\tilde{\phi}$, we obtain

$$|\pi_\eta(\phi) - \phi(\theta^*)| \leq L_{\tilde{\phi}}(\pi_\eta(f) - f^*) \leq L_{\tilde{\phi}}\Big(g^{-1}\Big(\frac{2\tilde{M}\eta}{2 - \tilde{L}\eta}\Big) + \frac{2\tilde{M}\eta}{2 - \tilde{L}\eta}\Big)$$

as desired. $\qquad\square$

We now prove Theorem 3.3.

*Proof of Theorem 3.3.* Consider the chain $\{\theta_k^{(\eta)}\}_{k \geq 0}$ starting from the stationary distirbution $\pi_\eta$. By display (15), it holds that

$$\mathbb{E}[\langle \nabla f(\theta_k^{(\eta)}), \theta_k^{(\eta)} - \theta^*\rangle] \leq C_2\eta\,,$$

where $C_2$ is a positive constant defined in Theorem 3.1. Note that $f$ is convex, this implies

$$0 \leq f(\theta_k^{(\eta)}) - f^* \leq \langle \nabla f(\theta_k^{(\eta)}), \theta_k^{(\eta)} - \theta^*\rangle\,.$$

Taking the expectation on both sides and combing this with the previous display gives

$$0 \leq \pi_\eta(f) - f^* \leq C_2\eta\,.$$

The desired result readily follows for the test function $\phi$ satisfying $\phi = \tilde{\phi} \circ f$ with the $L_{\tilde{\phi}}$-Lipschitz function $\tilde{\phi}$. $\qquad\square$

## B   Proposition B.1

**Proposition B.1.** *Let Assumptions 2.1, 2.2, and 3.1 hold. For $\theta^*$ denoting an arbitrary critical point of the objective function $f$, define the constants $\bar{L} := L(1 + \|\theta^*\|)$, and*

$$c_{L,\alpha} := \frac{\alpha - \sqrt{(\alpha^2 - (3L^2 + L_\xi)) \vee 0}}{3L^2 + L_\xi} \quad and \quad c_{L,\alpha}^\dagger := \frac{\alpha - \sqrt{(\alpha^2 - 16L_\dagger) \vee 0}}{64L_\dagger} \tag{16}$$

*with $L_\dagger := \bar{L}^2 + 16\Big(L_\xi^{3/4}\big(1 + (\beta/\alpha)^3\big) \vee L_\xi^{1/2}\big(1 + (\beta/\alpha)^2\big) \vee L_\xi\big(1 + (\beta/\alpha)^4\big)\Big)$. Then, for SGD iterates initialized at a point $\theta_0 \in \mathbb{R}^d$ and a step size satisfying $\eta < 1 \wedge \frac{1}{10L} \wedge c_{L,\alpha} \wedge c_{L,\alpha}^\dagger$, we have*

$$\mathbb{E}\big[\|\theta_k^{(\eta)} - \theta^*\|^4\big]^{1/2} \leq \rho^k \|\theta_0 - \theta^*\|^2 + D\,, \tag{17}$$

*where $\rho := \sqrt{1 - 2\alpha\eta + 32L_\dagger\eta^2} \in (0, 1)$. Consequently, for any $L_\phi$-Lipschitz test function $\phi$,*

$$\big|\pi_\eta(\phi) - \phi(\theta^*)\big| \leq L_\phi\sqrt{D}\,,$$

*where $D := \frac{64}{\alpha}\Big(\bar{L}^4 + L_\xi\big(1 + (\beta/\alpha)^4\big) + 512\bar{L}^6 + 23L_\xi^{3/2}\big(1 + (\beta/\alpha)^6\big)\Big)^{1/2}$*

$$\vee \frac{8}{\alpha}\Big(\beta + (\sqrt{\alpha} + 2L/\sqrt{\alpha})^2\|\theta^*\| + L\|\theta^*\| + 6\bar{L}^2 + 9L_\xi^{1/2}\big(1 + (\beta/\alpha)^2\big) + 16\Big).$$

The above theorem establishes that the SGD algorithm initialized far away from any critical point will converge (in the 4-th expectation) to the ball that contains all the first-order critical points exponentially fast. The first term in the upper bound (17) depends on the initialization, but decays to zero exponentially fast with the number of iterations, for a fixed step size. This convergence speed depends on the constant $\rho$ which depends on $\eta$, and the convergence of the SGD iterates to the stationary distribution becomes slower as $\eta \to 0$. The second term in the bound (17) is a constant independent of the iteration number as well as the step size, which serves as the squared radius of the ball that contains all the critical points plus an additional offset to account for the randomness in the SGD iterates. In other words, SGD algorithm initialized at any point and with any sufficiently small step size will find this ball of interest exponentially fast.

## C  Proofs for Section 4

### C.1  Verification of Assumptions for Example 4.1

**Asymptotic normality:**

*Proof of Lemma 4.1.*  The above objective has the following gradient

$$\nabla f(\theta) = \frac{1}{m} \sum_{i=1}^{m} \frac{\mathbf{x}_i(\langle \mathbf{x}_i, \theta \rangle - y_i)}{1 + (y_i - \langle \mathbf{x}_i, \theta \rangle)^2} + \lambda \theta.$$

Because $\|\nabla f(\theta)\| \leq \left(\lambda_{\max}(\frac{1}{m}\mathbf{X}^\top \mathbf{X}) + \lambda\right)\|\theta\| + \frac{1}{m}\|\mathbf{X}^\top \mathbf{y}\|$ by the triangle inequality and the fact that the denominator is lower bounded by 1, Assumption 2.1 holds. For Assumption 2.2, we write

$$\langle \nabla f(\theta), \theta \rangle = \frac{1}{m} \sum_{i=1}^{m} \frac{(\langle \mathbf{x}_i, \theta \rangle)^2 - y_i \langle \mathbf{x}_i, \theta \rangle}{1 + (y_i - \langle \mathbf{x}_i, \theta \rangle)^2} + \lambda \|\theta\|^2 \geq -\left\|\frac{1}{m}\mathbf{X}^\top \mathbf{y}\right\|\|\theta\| + \lambda \|\theta\|^2,$$

by Cauchy-Schwartz inequality. Next, using Young's inequality $-\left\|\frac{1}{m}\mathbf{X}^\top \mathbf{y}\right\|\|\theta\| \geq -\frac{1}{\lambda}\left\|\frac{1}{m}\mathbf{X}^\top \mathbf{y}\right\|^2 - \frac{\lambda}{4}\|\theta\|^2$, Assumption 2.2 holds for $\alpha = \lambda/4$ and $\beta = \frac{1}{\lambda}\left\|\frac{1}{m}\mathbf{X}^\top \mathbf{y}\right\|^2$. Finally, the gradient noise has finite 4-th moment with support on $\mathbb{R}^d$; thus, Assumption 2.3 is satisfied, and Theorem 2.1 is applicable. $\square$

**Bias:**

*Proof of Lemma 4.2.*  To verify assumptions, we compute the gradient and the Hessian respectively as

$$\nabla f(\theta) = \frac{\theta}{1 + \|\theta\|^2} + \lambda \theta, \qquad \text{and} \qquad \nabla^2 f(\theta) = \frac{I}{1 + \|\theta\|^2} - \frac{2\theta\theta^\top}{(1 + \|\theta\|^2)^2} + I\lambda,$$

with $I$ denoting the identity matrix. For small $\lambda$ the above function is clearly non-convex. To see this, choose $\lambda = 0.1, u = \theta/\|\theta\|$ and note that $\langle u, \nabla^2 f(\theta)u \rangle < 0$ whenever $1.5 \leq \|\theta\| \leq 2$. Also, note that

$$\|\nabla f(\theta)\|^2 = \|\theta\|^2 \left(\lambda + 1/(1 + \|\theta\|^2)\right)^2 \geq \frac{2\lambda^2}{1 + \lambda}\{f(\theta) - f(\theta^*)\}.$$

Thus, Assumption 3.3 is satisfied for $\gamma = \frac{2\lambda^2}{1+\lambda}$ and $g(x) = \gamma x^2$. Following the same steps in the regression setting, one can also verify Assumptions 2.1-2.2. Moreover, by definition of noise sequence (4), Assumption 3.1 is satisfied. Hence, Theorem 3.2 can be applied. $\square$

### C.2  Verification of Assumptions for Example 4.2

**Asymptotic normality:**

*Proof of Lemma 4.3.*  Indeed, it has the gradient

$$\nabla f(\theta) = -\frac{1}{m} \sum_{i=1}^{m} \frac{\mathbf{x}_i \left(y_i - \langle \mathbf{x}_i, \theta \rangle\right) e^{-(y_i - \langle \mathbf{x}_i, \theta \rangle)^2}}{\nu + e^{-(y_i - \langle \mathbf{x}_i, \theta \rangle)^2}} + \lambda \theta.$$

The triangle inequality yields

$$\|\nabla f(\theta)\| \leq \frac{1}{1 + \nu}\left\|\frac{1}{m}\mathbf{X}^\top \mathbf{y}\right\| + \left(\frac{1}{1 + \nu}\lambda_{\max}(\frac{1}{m}\mathbf{X}^\top \mathbf{X}) + \lambda\right)\|\theta\|,$$

which verifies Assumption 2.1. To verify the dissipativity assumption, we can write

$$\langle \nabla f(\theta),\, \theta \rangle = \langle -\tfrac{1}{m} \sum_{i=1}^{m} \frac{\mathbf{x}_i \big(y_i - \langle \mathbf{x}_i, \theta \rangle\big) e^{-(y_i - \langle \mathbf{x}_i, \theta \rangle)^2}}{\nu + e^{-(y_i - \langle \mathbf{x}_i, \theta \rangle)^2}} + \lambda\theta,\, \theta \rangle \geq -\tfrac{1}{1+\nu} \big\| \tfrac{1}{m} \mathbf{X}^\top \mathbf{y} \big\| \|\theta\| + \lambda \|\theta\|^2 \,.$$

The inequality follows from the triangle and Cauchy-Schwartz inequalities. Using Young's inequality, we obtain

$$-\frac{1}{1+\nu} \big\| \tfrac{1}{m} \mathbf{X}^\top \mathbf{y} \big\| \|\theta\| \geq -\frac{1}{\lambda(1+\nu)} \big\| \tfrac{1}{m} \mathbf{X}^\top \mathbf{y} \big\|^2 - \frac{\lambda}{4(1+\nu)} \|\theta\|^2,$$

which shows that the above function is dissipative for $\alpha = \lambda/2$ and $\beta = \frac{1}{2\lambda(1+\nu)^2} \big\| \tfrac{1}{m} \mathbf{X}^\top \mathbf{y} \big\|^2$; thus, Assumption 2.2 holds. $\qquad\square$

**Bias:**

*Proof of Lemma 4.4.* We write the gradient and the Hessian respectively, as

$$\nabla f(\theta) = \frac{\theta}{1 + \nu e^{\|\theta\|^2}} + \lambda\theta \qquad \text{and} \qquad \nabla^2 f(\theta) = \frac{I}{1 + \nu e^{\|\theta\|^2}} - \frac{2\nu e^{\|\theta\|^2}}{(1 + \nu e^{\|\theta\|^2})^2} \theta\theta^\top + \lambda I.$$

First, note that the Hessian can have negative eigenvalues for small values of $\lambda$. For example, for $\nu = 1$, $\lambda = 0.1$, and the unit direction $u = \theta/\|\theta\|$, we have $\langle u,\, \nabla^2 f(\theta) u \rangle < 0$ for $1 \leq \|\theta\|^2 \leq 2$; thus the function is non-convex. But we also have

$$\langle \nabla f(\theta),\, \theta \rangle = \|\theta\|^2 \Big( \lambda + 1/\big(1 + \nu e^{\|\theta\|^2}\big) \Big) \geq \Big( \lambda + 1/\big(1 + \nu e^{R^2}\big) \Big) \|\theta\|^2$$

for $\|\theta\| \leq R$ and $\langle \nabla f(\theta),\, \theta \rangle \geq \lambda \|\theta\|^2$ for $\|\theta\|^2 > R$; thus, Assumption 3.2 is satisfied for $\alpha = \lambda$, and any $\beta \geq 0$ and $g(x) = \Big( \lambda + 1/\big(1 + \nu e^{R^2}\big) \Big) x^2$. Following the same steps in the previous example, one can also verify Assumptions 2.1, 3.1; therefore, Theorem 3.1 follows. $\qquad\square$

# D   Simulation setting for constructing CIs

Here, we provide more details about constructing the CIs based on the three methods.

**Subsampling quantile:** For each trajectory, we first compute the means of subsamples for each trajectory using a rolling window of size 200. We then compute the 0.95 quantile of the empirical distribution of the absolute values of the differences between the rolling means and the mean of this trajectory. Employing the formula (4.4) and Corollary 4.2.1 in [89, Sections 4.2], we can construct the 95% confidence intervals accordingly.

**Subsampling var:** For each trajectory, we calculate the variances of subsamples for this trajectory using a rolling window of size 200. Combined with the mean of this trajectory, we then compute the 95% confidence interval.

**Long-run var:** For each trajectory, we compute the Newey-West estimate for the long-run variance of this trajectory using the **sandwich** package (version 3.0-1) in R 4.0.3 (function `lrvar()` with `type= "Newey-West"` ). Then, with the mean of this trajectory, we can compute the 95% confidence interval.