# OpenReview forum: "An Analysis of Constant Step Size SGD in the Non-convex Regime: Asymptotic Normality and Bias"
_NeurIPS.cc/2021/Conference — NeurIPS 2021 Poster_

### Official Review · Reviewer_HoaV · 2021-07-15

**Rating:** 6
**Confidence:** 3

**Summary:**

This paper studies the asymptotic normality and the bias for non-convex optimization problem with constant stepsize SGD algorithm. Relaxing previous strongly convex assumption, adopting techniques from Markov chains, they show that under some mild conditions, SGD with constant stepsize will quickly converge to a stationary distribution that is uniquely determined by the stepsize $\eta$. Also, they analyze the bias under various conditions, basically showing that the bias is also (positively) dependent of the stepsize $\eta$. Last, they provide thorough experiments in Regularized MLE and Regularized Blake-Zisserman MLE settings, showing that the numerical results match their theoretical prediction.

**Limitations And Societal Impact:**

Yes

**Main Review:**

Overall, this is a good and well-written paper. It studies a very important problem: how to quantify the uncertainty of SGD in the realistic non-convex and non-smooth setting. Also, its proof techniques may have independent interest. Different from previous work, this paper uses $V$-uniform ergodicity to show the existence and uniqueness of the stationary distribution (Proposition 2.1). Some detailed suggestions and comments are listed below and I would be happy if you can answer this questions.

1. In the application part, you list the Regularized MLE and Regularized Blake-Zisserman MLE. I am interesting that whether we can show that neural network also satisfies the conditions required to have the asymptotic normality. To me, with some regularizer or weight decay, at least shallow networks is hopeful to satisfy your conditions.

2. As you already mentioned, there is an interplay between the convergence to the stationary distribution and the bias term. If $\eta$ is large, then the convergence is quick. On the other hand, the bias will become large if $\eta$ is large. Could you provide a quantitive analysis to provide some suggestions on how to choose a good stepsize.

**Time Spent Reviewing:**

3 hours

---

> ### Author Response · Authors · 2021-08-10
> **Response to Reviewer HoaV**
>
> We thank the reviewer for their positive feedback.
>
> 1. We agree that a subset of $\ell_2$-regularized neural networks will satisfy the linear growth condition and the dissipativity assumption on the gradient of the objective. Therefore, if the noise model also satisfies our conditions (Assumption 2.3), then our asymptotic normality result should hold.
>
>     However, one should verify the condition on the noise with care, as some recent studies [1,2] show empirically that in neural networks, the gradient noise may exhibit heavy-tailed behavior, which may ultimately violate our noise assumption (Assumption 2.3). In the case of heavy-tailed noise, there are already results (under conditions even **stronger than strong convexity**) that SGD average is not asymptotically normal, but instead convergences weakly to an $\alpha$-stable distribution [3]. We are not aware of a similar result in the non-convex regime (or even in the strongly convex regime). Thus, extending our noise assumption to cover the heavy-tailed noise setup seems to be an important but very non-trivial step, which we leave for future work.
>
>
> 2. Indeed, in terms of $\eta$ there is an interplay between the convergence to the stationary distribution and the bias term. However, there is also an additional parameter $k$ (i.e., the number of iterations) at our disposal to control the overall error.
>
>     The choice of $\eta$ and the number of iterations $k$ together determine the overall error between the $k$-th SGD iterate and a critical point $\theta^*$. In order to determine the overall error after $k$-th iteration, to balance the bias and convergence, one needs to know how the parameter $\rho$, appearing in Proposition 2.1, which depends on $\eta$ (please also see lines 153-158). In fact, in our proofs, the precise dependency of $\rho$ on $\eta$ is characterized implicitly. This is done in order to guarantee that $\rho <1$, which is sufficient for ergodicity, which was needed to establish our CLT result.
>
>     We demonstrate the aforementioned interplay in a simplified model: $\text{error}(k,\eta) = \rho(\eta)^k + \text{bias}(\eta)$. Suppose $\rho(\eta) =1-\eta$ and $\text{bias}(\eta)=\eta$, then the error will be of order $(1-\eta)^k + \eta$, which is upper bounded by $e^{-\eta k/2} + \eta$. If we want $\text{error}(k,\eta)$ to be less than $\epsilon$, then we want $e^{-\eta k/2} + \eta = \epsilon/2 +\epsilon/2$. Hence, one typically sets $\eta = \epsilon/2$ and determines $k$ to be of order $\mathcal{O}(1/\epsilon \log(1/\epsilon))$. However notice that the choice of $\eta$ is determined entirely by the selected accuracy level $\epsilon$ and the bias term, i.e. solving for $\text{bias}(\eta) = \epsilon/2$, and does not depend on the converging term $\rho(\eta)^k$. This term, after solving $\rho(\eta)^k= \epsilon/2$ for $k$, determines the convergence rate.
>
>     An interesting scenario in which it is required to know $\rho(\eta)$ explicitly (from our proofs) to choose the step-size $\eta$ is when we pre-specify the number of iterations $k$, due to, say, budget constraints. In this case, we minimize the $\text{error}(k, \eta)$ for the step-size $\eta$ for a pre-specified value of $k$.
>
>
> We thank the reviewer for their stimulating questions, both of which will be discussed in the final version of our paper as important future work. We will be happy to discuss any follow-up questions.
>
> ---
>
> [1] Umut Simsekli, Levent Sagun, and Mert Gurbuzbalaban. "A tail-index analysis of stochastic gradient noise in deep neural networks." International Conference on Machine Learning, ICML, 2019.
>
> [2] Jingzhao Zhang, Sai Praneeth Karimireddy, Andreas Veit, Seungyeon Kim, Sashank J. Reddi, Sanjiv Kumar, and Suvrit Sra. "Why are adaptive methods good for attention models?." Advances in Neural Information Processing Systems, NeurIPS, 2020.
>
> [3] Hongjian Wang, Mert Gurbuzbalaban, Lingjiong Zhu, Umut Simsekli, and Murat Erdogdu, Convergence Rates of Stochastic Gradient Descent under Infinite Noise Variance, arXiv 2021

---

### Official Review · Reviewer_LBYf · 2021-07-16

**Rating:** 6
**Confidence:** 4

**Summary:**

In the paper, the authors establish an asymptotic normality result for the constant step size stochastic gradient descent (SGD) algorithm under the dissipative settings of non-convex and non-smooth objective functions.

**Ethical Concerns:**

I do not detect any ethical issues with the paper.

**Ethics Review Area:**

["I don’t know"]

**Limitations And Societal Impact:**

I do not see any foreseeable limitations and negative societal impact of this work.

**Main Review:**


In my opinion, the asymptotic results of the constant step size SGD presented in the paper under non-smooth and non-convex objective settings are of interest. Here are my comments on the paper:

(1) The dissipativity assumptions in Assumptions 2.2 or 3.2 are satisfied by also symmetric two components Gaussian mixtures, non-linear regression models, and informative non-response model (See references [1] and [2] that I mentioned below for more details).

(2) In Proposition 1, since $L$ and $L_{\xi}$ are not known in practice, what is the practical way to choose $\eta$ to satisfy the constraint in that proposition? Furthermore, do we have some explicit forms for the stationary distribution or simply just some approximation?

(3) In Theorem 3.1, I also have the same comment on the learning rate as the previous comment. We usually do not know $\alpha, \beta$, and the growth of function $g$ in localized dissipativity assumption. Furthermore, it seems that the upper bound of the expectation of $\|theta^{(\eta)} - \theta^{*}\|$ is not tight when $g(x) = x^{p}$ for $p > 2$. Can the authors clarify more this result?

(4) The results of Theorem 3.2 are not readable. I do not understand the meanings of constants $M$ and $m$ in that theorem. Furthermore, what is the implication of the results of Theorem 3.2 apart from the complicated constants?

(5) I think that in the localized dissipativity setting where $g(x) = x^{p}$ with $p > 2$, the CLT of constant step size SGD is not a normal distribution. In fact, it seems to be a Gaussian process. Can the authors provide more insight into this setting?

(6) There are a few relevant papers around the context of the paper that the authors may consider adding to the work:

(6.1) For the literature for the computational complexity and statistical accuracy of estimators in general non-convex settings (need not be locally strongly convex), the current work of [1] established the trade-off between the instability of general optimization algorithms and their statistical accuracy and computational efficiency.

(6.2) The dissipativity and local dissipativity also recently have been used to study the posterior convergence rates of parameters under Bayesian models [2].

--- References:

[1] N. Ho, R. Dwivedi,  K. Khamaru, M. J. Wainwright, M. I. Jordan, B. Yu. Instability, computational efficiency, and statistical accuracy. Arxiv preprint Arxiv 2005.11411, 2020.

[2] W. Mou, N. Ho, M. J. Wainwright, P. Bartlett, M. I. Jordan. A diffusion process perspective on the posterior contraction rates for parameters. Arxiv preprint Arxiv 1909.00966, 2019.

**Time Spent Reviewing:**

2 hours

---

> ### Author Response · Authors · 2021-08-10
> **Response to Reviewer LBYf**
>
> We thank the reviewer for their valuable feedback. Please find our point by point response below.
>
> - **(1)** We thank the reviewer for bringing these examples to our attention, which we think are of great relevance to our work. We will add a thorough discussion on these examples in the final version of our paper, pointing to the references mentioned by the reviewer.
>
> - **(2)** For some statistical models, the constants $L$ and $L_\xi$ can be (conservatively) bounded based on the properties of the data generating distribution. We demonstrated this in Section 4 (Examples and Numerical Experiments) on two non-convex objectives (see Lemmas~4.1-4). Even conservative bounds on these parameters suffice to choose a theoretically feasible step-size.
>
>     In a model-agnostic setup (for example, prediction with deep neural networks), a popular heuristic which is often successfully employed in practice is to simply choose a sufficiently small step-size and track the descent of the optimization procedure. Since our condition on the step-size $\eta$ is guaranteed to be satisfied for sufficiently small $\eta$, one can i) choose a step-size $\eta$ (which is a small universal constant), ii) run SGD for a few iterations, iii) if the procedure does not provide descent on average, update the step-size as $\eta \leftarrow \eta/2$.
>
>     Characterizing the stationary distribution of SGD is an extremely interesting and challenging problem, beyond the scope of the current submission. Except for a few special/simple cases (e.g. quadratic minimization problems [MLWBJ20]), we are not aware of any results explicitly characterizing the stationary distribution of SGD in the general setting.
>
>
>
> - **(3)**  Similar to the previous case,  for some specific statistical models, we can (conservatively)  obtain $\alpha$ and $\beta$ such that the dissipativity condition is satisfied, both of which would depend on the properties of the data generating distribution. We again refer to Section 4 (Examples and Numerical Experiments) and Lemmas~4.1-4 where we derive these constants for specific problems. If the model is not available to the practitioner, to find a feasible step-size, one can follow the procedure suggested in the above item.
>
>      The convex function $g$ characterizes the local growth of the objective function $f$ around the unique minimizer $\theta^*$. If we assume $g(x)=x^p, p \ge 1,$  it then follows from Theorem 3.1 that $\mathbb{E}[\|\theta^{(\eta)}-\theta^*\|]\lesssim  \eta + \eta^{1/p}.$
> When $p$ increases, the objective function tends to be more flat around the minimizer, which yields the slower convergence rate for the bias since the term $\eta^{1/p}$ dominates the right hand side. We ask the reviewer to kindly follow up on their point if there are additional questions.
>
> - **(4)** The important take-away message from Theorem 3.2 (and Theorem 3.1 and 3.3) is that under additional assumptions regulating the local behavior of the objective around its critical points, the bias of constant step-size SGD could be controlled  by the step-size parameter $\eta$; smaller $\eta$ yields smaller bias.
>
>     Theorem 3.2 in particular quantifies the bias under the generalized Lojasiewicz condition, and this result implies that a smaller step size gives smaller bias after the SGD chain converges to the stationary distribution. The explicit form of $M$ and $m$ provides the quantitative insight on that how do the constants $L,\tilde L,$ $L_\xi,\alpha,\beta$ in the assumptions enter the bound. For example when $g(x)=x$, the order of the bias is $\pi_\eta (f) - f(\theta^*) \lesssim \eta$; however, we also provide the upper bound with explicit constants depending on problem parameters, i.e. $\pi_\eta (f) - f(\theta^*) \le 4M\eta / (2-\tilde{L}\eta)$ where $M$ is defined in the theorem statement and it depends on $m$.
>
>
> - **(5)** We note that the function $g$ characterizes the local growth of the objective. The CLT result (Theorem 2.1) does not depend on the local behavior of the objective, and it holds under Assumptions 2.1, 2.2 and 2.3.  In other words, the **local** part of the localized dissipativity condition does not change the asymptotic normality, thus the CLT still holds possibly with a different $\pi_\eta$. We kindly request the reviewer to follow up on their point in the discussion period, if there are additional questions.
>
> - **(6)** We thank the reviewer for pointing us to those papers. A thorough discussion on these papers will be included in the final version of our work.
>
> We would be happy to clarify any further concerns/questions in the discussion period.
>
> ---
>
> [MLWBJ20] Wenlong Mou, Chris Junchi Li, Martin J. Wainwright, Peter Bartlett, Michael I. Jordan. "On Linear Stochastic Approximation: Fine-grained Polyak-Ruppert and Non-Asymptotic Concentration", 2020

---

> > ### Comment · Reviewer_LBYf · 2021-08-29
> > **Thank you**
> >
> > I would like to thank the authors for the detailed rebuttal. They already addressed my concerns and I am happy with the current paper. The authors can try to incorporate my comments and  other reviewers' comments in their revision.

---

### Official Review · Reviewer_8kWx · 2021-07-17

**Rating:** 8
**Confidence:** 4

**Summary:**

The paper provides CLT and bias control for constant step-size stochastic gradient descent in the non-convex case. The general theory is applied to several specific examples.

**Main Review:**

The paper presents an interesting results on SGD algorithm with constant stepsizes. Under reasonable assumptions, the authors study the limiting behaviour of SGD output. It is well known, that constant stepsize SGD produces a biased stationary measure. The author quantifies this phenomenon. First, they show CLT for biased sequence and next provide bounds on the bias. The main result is applied to several examples. In my opinion, the paper provides an important and interesting contribution to the theory of SGD with constant stepsize.

**Time Spent Reviewing:**

10h

---

> ### Author Response · Authors · 2021-08-10
> **Response to Reviewer 8kWx**
>
> We thank the reviewer for their positive feedback. We would be happy to clarify any concerns or answer any questions that may come up during the discussion period.

---

### Official Review · Reviewer_iHb7 · 2021-07-17

**Rating:** 4
**Confidence:** 4

**Summary:**

This paper provides a theoretical analysis of the average version of SGD iterates, and shows that the averaged version of SGD iterates is asymptotically normally distributed around the expected value of their unique invariant distribution, under a generalized condition of dissipativity.

**Limitations And Societal Impact:**

From my perspective, the main limitation is the novelty. The analysis is analogous to the DDB19, but just extends the analysis to some generalization of the PL condition. The experiments also did not justify the nonconvex settings, nor surprising.

**Main Review:**

This paper generalizes the DDB19 to a more generalized condition than the PL condition. Though the authors claim the non-convex regime, the main contribution falls in the use of PL condition and the analysis is analogous to DDB19.

**Time Spent Reviewing:**

5

---

> ### Author Response · Authors · 2021-08-10
> **Response to Reviewer iHb7**
>
> We thank the reviewer for their comments. To address reviewer's concerns, we provide a thorough comparison with [DDB19] below. We kindly ask the reviewer to reassess our work in light of our response.
>
> We note that [DDB19] studies the constant step size SGD under a **strong convexity** assumption on the objective function which is also assumed to be 5 times continuously differentiable with **2nd to 5th uniformly bounded derivatives**.  In contrast, our results hold for **non-convex and non-smooth** objectives as long as they satisfy significantly weaker/general conditions as given in Assumptions 2.1, 2.2 and 2.3 in our paper.
>
> In particular, we emphasize that [DDB19] **does not** have a central limit theorem (CLT) result which we consider as one of our key contributions. Moreover, their convergence analysis relies heavily on strong convexity; thus our results are by no means straightforward to establish given the work of [DDB19]. As mentioned in lines 145-146, different from [DDB19], our proof relies on $V$-uniform ergodicity and general local growth conditions.
>
> The following table summarizes the main differences between [DDB19] and our work.
>
> |                 |                 |                |
> |--------------|-----------------|----------------|
> |              | **DDB19** | **Our work**       |
> | **Objective function** |  i) strongly convex            |  i) Non-convex     |
> |            | ii) 1st, 2nd, 3rd, 4th-order smoothness | ii) non-smooth |
> |            | iii) 5 times differentiable | iii) single time differentiable   |
> | **CLT for SGD average** | No | Yes  |
> | **Convergence analysis relies on** | global strong convexity | i) V-uniform ergodicity (dissipativity) |
> |  |  | ii) Bias is controlled under arbitrary local growth around the minimizer which is characterized by a function $g$ that leads to different convergence behavior (PL condition is a special case) |
> |                 |                 |                |
>
> ---
>
> We emphasize that **none** of our contributions primarily depend on the PL condition in particular. We would like to point out that our main contributions include
> - (A) the existence and uniqueness of the stationary distribution of the constant step-size SGD (Proposition 2.1),
> - (B) asymptotic normality (CLT) of the averaged SGD iterates (Theorem 2.1),
> - (C) bias control with the localized  dissipativity condition (Theorem 3.1),
> - (D) bias control with the generalized Lojasiewicz condition (Theorem 3.2),
> - (E) bias control under (non-strong) convexity (Theorem 3.3).
>
> Specifically, contributions (A) and (B) require only dissipativity (Assumption 2.2) and linear tail growth on the gradient (Assumption 2.1), both are **tail growth** assumptions not related to (generalized) PL-type conditions. Out of the contributions in (C), note that the convex function $g:[0,\infty]\to [0,\infty]$ characterizes the local growth behavior of the objective function, and the PL condition that the reviewer mentions is **implied** in the special case (with $g(x) = x^2$).  Similarly in contribution (D), our assumption reduces to the PL condition **locally** in the special case $g(x)=x$, and reduces to the generalized PL condition **locally** in the special case $g(x)=x^\kappa$.
>
> This is summarized in the following table.
>
> | *Our results* | **PL inequality** | **Generalized PL inequality** |
> |--------------|-----------------|----------------|
> | **Prop 2.1 (Ergodicity)** | Not relevant | Not relevant  |
> | **Thm 2.1 (CLT)** |  Not relevant |  Not relevant |
> |  **Thm 3.1 (localized dissipativity)** | Thm holds for all convex $g:[0,\infty]\to[0,\infty]$ characterizing local growth. Condition **implies** PL in the special case $g(x)=x^2$ | Condition **implies** generalized PL in the special case $g(x)=x^\kappa$ |
> |  **Thm 3.2 (generalized Lojasiewicz)** | Thm holds for all convex $g:[0,\infty]\to[0,\infty]$ characterizing local growth. Condition **locally** reduces to PL in the special case $g(x)=x$ | Condition **locally** reduces to generalized PL in the special case $g(x)=x^\kappa$   |
> | **Thm 3.3 (convexity)** | Not needed/does not imply | Not needed/does not imply |
> |                 |                 |                |
>
> Finally, **both examples** discussed in Section 4 (*Examples and Numerical Studies*) **are non-convex**, and what we demonstrated in the figures cannot be rigorously explained by any prior work. We prove their non-convexity in Lemmas 4.2 and 4.4, proofs of which were moved to appendix due to the space constraints. We will move these proofs to main text in the final version.
>
> We would be happy to clarify any further concerns/questions in the discussion period.
>
> ---
> [DDB19] Dieuleveut, Aymeric and Durmus, Alain and Bach, Francis. "Bridging the gap between constant step size stochastic gradient descent and markov chains", The Annals of Statistics, 2020

---

### Decision · Program_Chairs · 2021-09-27

**Decision:**

Accept (Poster)

**Comment:**

This paper studies the asymptotic normality and the bias for constant step size stochastic gradient descent (SGD) algorithm in the non-convex and non-smooth setting. It shows that if the non-convex and non-smooth objective function satisfies a dissipativity property, the average of SGD iterates is asymptotically normally distributed around the expected value of their unique invariant distribution. It also characterizes the bias between this expected value and the critical points of the objective function under different local regularity conditions. The paper is well-written and studies an important problem. The authors are encouraged to incorporate the feedback from the reviewers in the revision.